# Continual Pre-training of MoEs: How robust is your router?

**Benjamin Thérien**[1,2,5]                                  *benjamin.therien@mila.quebec*
**Charles-Étienne Joseph**[5]                          *charlesetienne.joseph@capitalone.com*
**Zain Sarwar**[4]                                              *zsarwar@uchicago.edu*
**Ashwinee Panda**[5]                                       *ashwinee@princeton.edu*
**Anirban Das**[5]                                             *anirban.das3@capitalone.com*
**Shi-Xiong Zhang**[5]                                      *shixiong.zhang@capitalone.com*
**Stephen Rawls**[5]                                         *stephen.rawls@capitalone.com*
**Sambit Sahu**[5]                                            *sambit.sahu@capitalone.com*
**Eugene Belilovsky**[2,3]                                 *eugene.belilovsky@concordia.ca*
**Irina Rish**[1,2]                                              *irina.rish@mila.quebec*

[1] *Université de Montréal, Montréal, Canada*
[2] *Mila – Quebec AI Institute, Montréal, Canada*
[3] *Concordia University, Montréal, Canada*
[4] *University of Chicago, Chicago, USA*
[5] *Capital One, New York, NY, USA*

**Reviewed on OpenReview:** *https://openreview.net/forum?id=dR7C1K71Rs*

## Abstract

Sparsely-activated Mixture of Experts (MoE) transformers are promising architectures for foundation models. Compared to dense transformers that require the same amount of floating-point operations (FLOPs) per forward pass, MoEs benefit from improved sample efficiency at training time and achieve much stronger performance. Many closed-source and open-source frontier language models have thus adopted an MoE architecture. Naturally, practitioners will want to extend the capabilities of these models with large amounts of newly collected data without completely re-training them. Prior work has shown that a simple combination of replay, learning rate re-warming, and re-decaying can enable the continual pre-training (CPT) of dense decoder-only transformers with minimal performance degradation compared to full re-training. In the case of decoder-only MoE transformers, however, it is unclear how the routing algorithm will impact continual pre-training performance: 1) *do the MoE transformer's routers exacerbate forgetting relative to a dense model?*; 2) *do the routers maintain a balanced load on previous distributions after CPT?*; 3) *are the same strategies applied to dense models sufficient to continually pre-train MoE LLMs?* In what follows, we conduct a large-scale study training a 500M parameter dense transformer and four 500M-active/2B-total parameter MoE transformers, following the Switch Transformer architecture and a granular DeepSeek-inspired architecture. Each model is trained for 600B tokens. Our results establish a surprising robustness to distribution shifts for MoEs using both Sinkhorn-Balanced and Z-and-Aux-loss-balanced routing algorithms, even in MoEs continually pre-trained without replay. Moreover, we show that MoE LLMs maintain their sample efficiency (relative to a FLOP-matched dense model) during CPT and that they can match the performance of a fully re-trained MoE at a fraction of the cost.

## 1 Introduction

Sparsely-activated MoE transformers achieve significantly stronger performance than FLOP-matched dense models (e.g., dense models requiring the same amount of floating point operations (FLOPs) per forward pass). This is particularly advantageous in today's foundation model lifecycle, where a model spends the majority of its lifetime FLOPs being inferenced. Many closed-source and open-source frontier language models have thus adopted an MoE architecture (Dai et al., 2024; DeepSeek-AI et al., 2024; Jiang et al., 2024; Abdin et al., 2024;

DeepSeek-AI et al., 2025b;a). Given the clear advantages of MoEs over dense transformers, practitioners will certainly want to update MoEs on new data as is currently done for dense transformers.

Continual pre-training with replay, learning rate re-warming, and re-decaying has been shown to be a simple but effective solution for updating pre-trained dense autoregressive transformers on large amounts of new data (Ibrahim et al., 2024; Gupta et al., 2023; Parmar et al., 2024) and is competitive with full re-training (Ibrahim et al., 2024), while being much cheaper. An open question is whether the same strategies are sufficient to continually pre-train MoE LLMs? MoE pre-training has been notoriously difficult due to instabilities introduced by the routing algorithm and the need to maintain a balanced load across experts (Lepikhin et al., 2021; Shazeer et al., 2017; Fedus et al., 2022; Zoph et al., 2022). During continual pre-training, these challenges may be exacerbated by the distribution shift.

Without proper care, MoE transformers learn greedy routing strategies that overutilize certain experts, leading to poorer downstream performance and poorer accelerator utilization. During MoE pre-training, load balancing strategies are used to prevent such negative outcomes (Fedus et al., 2022; Zoph et al., 2022; Clark et al., 2022; Anthony et al., 2024; Dai et al., 2024). However, the load-balancing algorithms used in SOTA MoEs were not specifically designed for the non-IID data distributions encountered during continual pre-training. Adapting the router's decisions to a new distribution during CPT may compromise the balanced load on previous distributions, potentially leading to exacerbated forgetting and poor accelerator utilization. Avoiding these failure modes is critical for successfully updating MoE foundation models without the need for full re-training, but the continual pre-training of MoEs has not yet been thoroughly studied in the literature.

In this work, we fill the gap by providing a systematic study of MoE continual pre-training. Specifically, we select two popular routing algorithms and two popular MoE architectures used in state-of-the-art existing work (Dai et al., 2024; Muennighoff et al., 2024; Fedus et al., 2022; Clark et al., 2022) to yield four different MoEs for our study. We then pre-train each MoE language model on 400B tokens of FineWeb and continually pre-train them on 200B tokens of code data and German web crawl data. Taking the strongest MoE architecture, we compare its performance to full re-training baseline on both datasets. Our contributions can be summarized as follows.

- We establish the effect of replay and infinite learning rate (LR) schedules on the forgetting and routing imbalance dynamics of MoE transformer LMs during CPT.

- We demonstrate that a Penalty-Balanced (e.g. with Z and Aux loss) MoE following the DeepSeek architecture can successfully match the performance of a full re-training baseline, at a fraction of the cost.

- We show that MoEs using either Penalty-Balanced or Sinkhorn-balanced routing algorithms are *surprisingly* robust to distribution shifts in terms of 1) language modeling performance, 2) evaluation benchmarks, and 3) maximum routing imbalance.

- We provide a comprehensive analysis of how routing decisions change during continual pre-training that provides insight into how MoEs adapt to new distributions and forget previous ones.

## 2 Background

This section provides a concise summary of the relevant background for our study. A more detailed version can be found in Section A of the appendix.

**Continual Pre-training of LLMs.** Continual pre-training (CPT) extends pre-training to one or more new distributions. Concretely, continual pre-training occurs when a model is trained on a sequence of datasets $\mathcal{D}_0, \mathcal{D}_1, \ldots, \mathcal{D}_N$ with different distributions, $N \geq 2$, and each dataset is sufficiently large (e.g., $> 100B$ tokens in the case of language). Recently, Ibrahim et al. (2024) established that CPT LLMs can match the performance of full re-training by simply repeating the pre-training schedule (cosine annealing) for CPT and replaying previous data. However, if one has control over the initial pre-training, it is possible to further improve CPT by using an infinite learning rate schedule (Janson et al., 2025).

**Sparsely-activated MoE transformers** differ from their dense counterparts by dynamically routing tokens in a sequence to different sets of parameters, called experts. Algorithms for dynamically selecting among

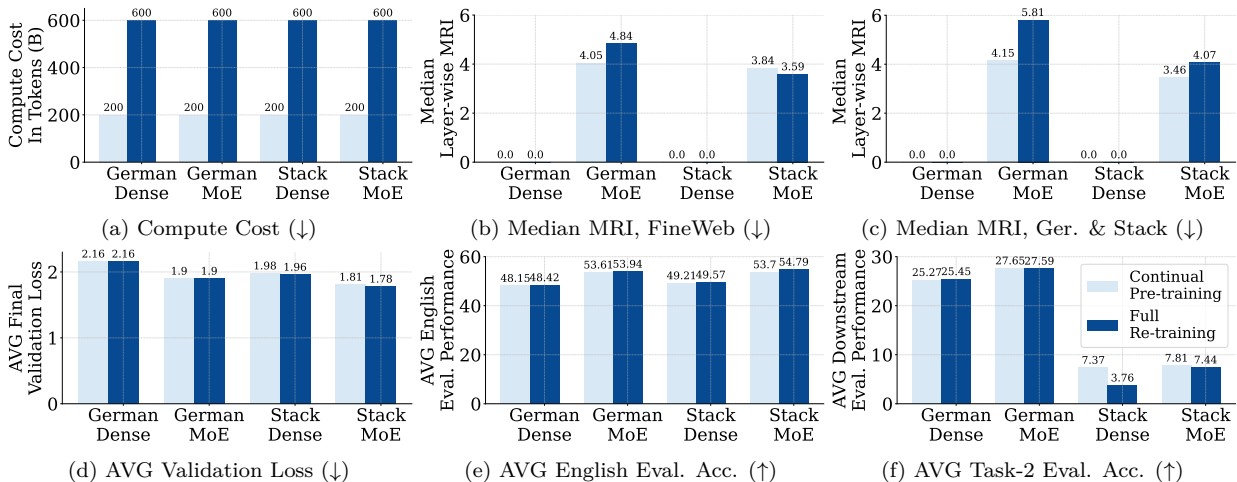

Figure 1: **Continually pre-trained (CPT) MoEs match the performance of full re-training across two dataset transitions: 400B English→ 200B German (40% replay) and 400B English→ 200B Stack (30% replay).** We compare the performance of a fully re-trained (e.g. trained on the union of 400B English and 200B stack or 200B German) Penalty-Balanced Top-$k$ MoE and dense baseline, to their CPT counterparts. Despite incurring only a third of the substantial full-retraining cost, the CPT MoEs match the performance of the fully re-trained models, even achieving improvements in median Maximum Routing Imbalance (MRI) in some cases. This shows that MoEs have CPT abilities on par with dense transformers. Note that subfigures (b), (c), and (f) evaluate German and Stack models on different datasets which correspond to their training domain.

experts, known as routing algorithms, are therefore central to MoEs and may be non-trivially affected by distribution shifts. In this work, we focus on studying two prominent Top-$k$ routing algorithms from recent state-of-the-art (SOTA) works: **Penalty-Balanced Top-$k$** (PBT$k$) routing (Shazeer et al., 2017; Dai et al., 2024; Zoph et al., 2022; Fedus et al., 2022) and **Sinkhorn-Balanced Top-$k$** (SBT$k$) routing (Clark et al., 2022; Anthony et al., 2024). PBT$k$ methods encourage a balanced load across experts by adding a penalty term (called *Aux Loss*) to the overall loss. Several recent SOTA works (Dai et al., 2024; Team, 2024) also include a second term, *z-loss*, to penalize large-magnitude router logits, which has been shown to promote stability (Zoph et al., 2022). In our experiments, we combine both losses and refer to the method as PBT$k$ routing. SBT$k$ methods cast the assignment of tokens to experts as a linear assignment problem (Clark et al., 2022). The Sinkhorn-knopp algorithm (Knopp & Sinkhorn, 1967) provides an approximate solution to this problem, which can be efficiently computed on GPUs and can be sped up by choosing a favourable initial condition (Anthony et al., 2024). We refer to Sinkhorn routing using the initialization of Anthony et al. (2024) as SBT$k$ in this work.

## 3 Related work

This section provides a review of the most relevant literature, which we complement with a more comprehensive review in Section B of the appendix.

**Continual Pre-training of Dense Foundation Models** Several existing works study continual learning in settings relevant to CPT, finding that self-supervised pre-training benefits from reduced forgetting (Cossu et al., 2022; Davari et al., 2022), that pre-trained models forget less than their randomly initialized counterparts (Mehta et al., 2023), that forgetting improves as model scale is increased (Ramasesh et al., 2022), and that wider models tend to forget less than deeper models (Mirzadeh et al., 2022). In the context of large-scale CPT of LLMs, Gupta et al. (2023) highlights the importance of re-warming the learning rate when models are pre-trained from a checkpoint that has been decayed to a small learning rate. Following up on their work, Ibrahim et al. (2024) establishes the effectiveness of learning rate re-warming, LR re-decaying, and replay for large-scale CPT of LLMs.

**Continual Pre-training of MoE LLMs.** To the best of our knowledge, only a single work exists exploring the large-scale continual pre-training of MoEs LLMs, while the majority of the literature focuses on upcycling or growing MoEs for continual pre-training. In a concurrent pre-print DeepSeek-CoderV2 (DeepSeek-AI et al., 2024), shows that they can continue from a checkpoint the training of a MoE LLM. However, this is only shown for one instance and the analysis of the MoE routing behavior is not discussed. Furthermore, there is no comparison to a FLOP-matched dense model, making it challenging to assess whether the sample efficiency of MoE LLMs is maintained during continual pre-training. Continual pre-training methods for MoEs that are less related to our work generally focus on fine-tuning MoE LLMs on small amounts of data (Wang et al., 2024c) or growing MoEs (Komatsuzaki et al., 2023; Zhu et al., 2024; Sukhbaatar et al., 2024; Gritsch et al., 2024).

## 4 Method & Empirical Study

Given our goal of studying the large-scale continual pre-training of MoE LLMs, our main methodological contribution involves identifying practically relevant MoE architectures to study, appropriately combining them with SOTA CPT techniques (e.g. (Ibrahim et al., 2024)), and providing succinct guidelines for continual MoE pre-training derived from our empirical results. In the following section, we will describe the key design choices we made when constructing our study w.r.t. MoE architectures, datasets, and CPT techniques and introduce a new metric for measuring latency in MoEs. Guidelines for the CPT of MoEs will be outlined in the results section.

### 4.1 Selected architectures for our study

**FLOP-matched Dense Baseline.** We select a 24 layer 570M parameter dense decoder-only transformer following the Llama3 architecture (except we use GeLU activations) and using the Llama3 tokenizer (Dubey et al., 2024) (see Sec. F for details).

**Granular MoEs.** Given the recent popularity and strong performance of DeepSeek MoEs (DeepSeek-AI et al., 2024; Dai et al., 2024; DeepSeek-AI et al., 2025b;a), we include an MoE architecture that activates multiple granular experts and a shared expert. Specifically, each granular MoE has $E = 31$ total routed experts, $K = 3$ active experts, and 1 shared expert. This model follows the same Llama3 architecture as the dense model described above. Notably, its experts are GEGLU FFNs with an intermediate size that is $\frac{1}{4}$ the size used in the dense model. We train two granular MoEs utilizing the Penalty-Balanced and Sinkhorn-Balanced Top-$k$ routing algorithms, respectively. We do not drop tokens.

**Switch MoEs.** Given the historical use of full-sized FFNs in MoEs, our study also includes an architecture similar to (Jiang et al., 2024; Fedus et al., 2022) with full-sized experts and no shared expert. We refer to these as Switch MoEs and also train two utilizing the Penalty-Balanced and Sinkhorn-Balanced routing algorithms, respectively. Each switch MoE has $E = 8$ total routed experts, $K = 1$ active experts, and no shared expert. This model follows the same Llama3 architecture as the dense model described above. Notably, its experts are GEGLU FFNs of the same size as the dense model's FFNs. We do not drop tokens.

### 4.2 Continual pre-training strategy and Datasets

To initially pre-train and subsequently continually pre-train our models, we use three datasets: FineWeb (Penedo et al., 2024), the Stack (Kocetkov et al., 2023), and German Common Crawl (Abadji et al., 2022). We initially pre-train all models on FineWeb for 400B tokens (task 1) to mimic open and closed source models often pre-trained on large-scale web-scraped English data. Subsequently, we continually pre-train these base models on 200B tokens of Code or German data (task 2) using infinite learning rate schedules and replay (30% & 40%, respectively) to mitigate forgetting. We select large amounts of replay for full continual pre-training following previous SOTA work (DeepSeek-AI et al., 2024), but show the effect of modifying the replay percentage in section 5.1. We chose distribution shifts to multilingual and code data as they represent stark distribution shifts from the English pre-training data, while being realistic (e.g., the Llama3 tokenizer is still viable for these domains). It should be noted that our total training budget of 600B tokens falls strictly in the overtraining regime with respect to a chinchilla optimal token budget for dense models (Hoffmann et al., 2022). For the Dense baseline, this corresponds to overtraining to roughly 40X the chinchilla optimal recommendation, while for the MoEs, this corresponds to roughly 10X the compute optimal amount for a 2B parameter dense model. For comparison, DeepSeekV3 (DeepSeek-AI et al., 2025b) trains to about

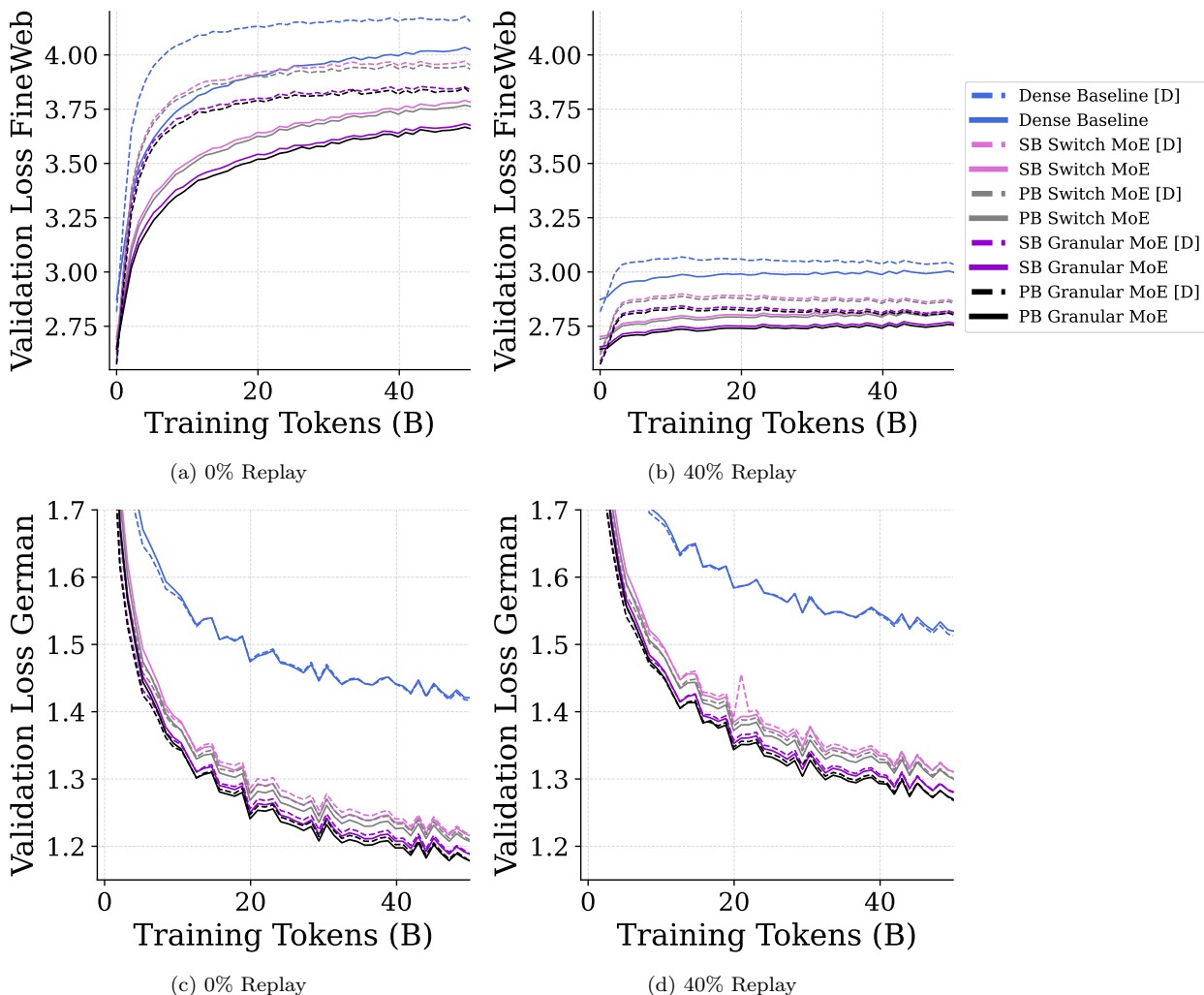

Figure 2: **Ablating replay and decay strategy during continual pre-training on German data.** We CPT MoEs and a dense baseline from fully-decayed checkpoints (dotted curves, [D]) or a non-decayed checkpoint (full curves). The figures report the performance on task 1 (FineWeb) and task 2 (German) while CPT on task 2. We observe that adaptation to task 2 is similar within an architecture for both checkpoints, that CPT from a non-decayed checkpoint improves forgetting, and that replay mitigates forgetting.

1.1X the Chinchilla optimal ratio for a 671B dense model and the popular Qwen3 series of models reaches chinchilla-optimal training multipliers of 7.66-58.06X for the MoEs and 54.55-225X for the larger (8B +) dense models. These models are frequently used as a starting point for continual pre-training, showing how our 40x chinchilla-optimal for dense and 10x chinchilla-optimal for MoEs is representative of real application settings.

**Compute Equivalent Replay** Replaying previous data has been a longstanding tool for mitigating catastrophic forgetting (Wang et al., 2024b). In our experiments, we replay previously seen data for this purpose, designating any model using the technique with the suffix "$X\%$ Replay". Here, $X$ represents the percentage of samples in a given batch that were replayed from the previous distribution. To match compute across different replay budgets, we do not increase the token budget when increasing the amount of replay. Instead, we decrease the amount of new data seen during CPT.

### 4.3 Training details

All models in our study (except re-training baselines) were pre-trained for $192,720$ gradient descent steps using a batch size of 1024, a sequence length of 2048, the AdamW optimizer, and the *Cosine Inf* schedule (Ibrahim et al., 2024). For continual pre-training, each model in the main study follows a *Cosine Inf* schedule resuming from the non-decayed checkpoint, while some models in the ablation section were continually pre-trained from decayed checkpoints following a cosine decay schedule (e.g., replicating the setting from Ibrahim et al. (2024)). We continually pre-train the models for $95,370$ gradient descent steps using the same batch size and sequence length as during pre-training. Each model was trained across 64 A100 GPUs using data parallelism and zero-1 (Rajbhandari et al., 2020). To accelerate the dropless MoE forward pass we use the Megablocks kernel (Gale et al., 2023). More details of the exact schedules used for each experiment are provided in section F of the appendix. Additionally, figure 22 illustrates learning rate schedules used for (a) pre-training, (b) continual pre-training, (c) full re-training, and (d) rewarming ablation of Sec. 5.1.

### 4.4 Maximum Routing Imbalance: A proxy for worst-case latency in MoEs

While performance is one important axis of robustness to distribution shifts, maintaining a balanced load across experts is just as important for MoE foundation models. Without a balanced load, MoE transformers inferenced using expert parallelism without token dropping (e.g., as is done for SOTA models (DeepSeek-AI et al., 2025b; Zhao et al., 2025)) could be bottlenecked by the speed of a single accelerator that receives all the tokens, leading to underutilization of the hardware, lower throughput, and higher costs. To quantitatively assess the effect of distribution shift on load balance, we propose the maximum routing imbalance (MRI): the largest proportion of tokens routed to a single expert in a given MoE layer. Concretely, the MRI at a training iteration $t$ and MoE layer $j$ is defined as

$$\text{MRI}(t, j) := \max_{i \in [1, ..., E]} \left[ \frac{\sum_{x \in B} \mathbb{1}\{i \in I_k(x)\}}{|B|} \right]. \tag{1}$$

Where $B$ is a set containing all tokens in a given batch, $\mathbb{1}$ is the indicator function, $E$ is the number of routed experts, and $k$ is the number of active experts. *Since latency increases with computation, and, in an MoE layer, the computation required by a given device increases with the load of experts on that device, then MRI calculated with respect to routing decisions on a distribution is a proxy for the worst case latency of an MoE layer on the distribution.* We will use the MRI throughout the following sections to measure the effect of algorithmic changes to continual pre-training on routing imbalance. In figure 3 we report the maximum MRI across all layers in the MoE (e.g., $\max_{j \in [L]} \text{MRI}(t, j)$, where L is the number of layers) at training iterations immediately preceding and immediately after the distribution shift. In figure 4 we report the MRI at each layer of our MoE transformers before and after continual pre-training.

**MRI v.s. Latency** While MRI does not report latency, it is a faithful behavioural metric that can be used as input to a latency model for estimating the latter. Unlike latency, which will always depend on specific hardware and implementation, MRI is independent of these considerations and is ultimately more comparable across different deployments.

## 5 Results

### 5.1 Ablating replay (%) and the checkpoint used for CPT

In the following section, we ablate the replay percentage used during continual pre-training and consider continually pre-training from two distinct checkpoints: a checkpoint that (a) was decayed to $\eta_{min}$ during pre-training (the case for most open-source MoEs) or (b) followed an infinite learning rate schedule and was not decayed (the ideal case achievable when one has control over the pre-training phase). Models in group (a) are continually pre-trained following a linear warmup and cosine decay schedule that rewarms the learning rate to $\eta_{max}$ before re-decaying it (e.g., as in Ibrahim et al. (2024)), while the models in group (b) are continually pre-trained starting from $\eta_{const}$ following an infinite LR schedule (exact values are provided in Sec. F).

**Validation Loss.** Figure 2 reports the validation loss for these models during the first 50B tokens of continual pre-training. While we only show the first 50B tokens due to resource constraints, the schedules were set to

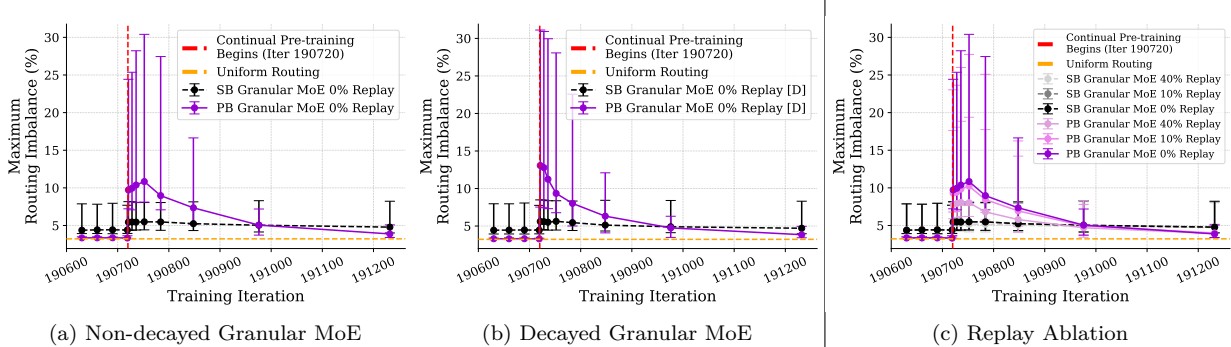

Figure 3: **FineWeb → German CPT checkpoint and replay ablation.** We report the median Maximum Routing Imbalance (MRI) across MoE layers with min/max error bars. Sinkhorn-Balanced (SBT$k$) MoEs show a slight MRI increase during distribution shift, while PBT$k$ MoEs experience a brief spike before recovering to balanced MRI levels below SBT$k$, which approach the uniform baseline. The uniform routing baseline (orange line) corresponds to the case where each expert across all layers receives the same number of tokens; thus, it represents perfect balance.

decay at 200B tokens, mimicking the start of a longer continual pre-training. Subfigures (a) and (c) show forgetting and adaptation plots using 0% replay, while subfigures (b) and (d) show analogous plots using 40% replay. We observe that as the percentage of replay is increased, the forgetting as measured by FineWeb validation loss is mitigated, while the adaptation to the downstream dataset is harmed. Turning our attention to the checkpoints used, we observe that, for all replay values and all models, using non-decayed checkpoints improves forgetting on the FineWeb without compromising adaptation. These results show that, similar to dense transformers, MoEs can tradeoff forgetting for adaptation with replay and benefit from infinite LR schedules

**Routing Imbalance.** Figure 3 (a,b) reports median MRI computed across all transformer blocks of SBT$k$ and PBT$k$ Granular MoEs during CPT with 0% replay, subfigure (c) reports results across different replay percentages and results for switch MoEs are reported in Figures 19 and 18 of the appendix. These figures precisely study the distribution shift by reporting the MRI immediately before and after the transition from English and German data. We observe that SBT$k$ MoEs are consistently robust to the distribution shift, showing only a small increase in MRI across different replay percentages and for decayed and non-decayed checkpoints alike. This is likely attributable to the explicit balancing step in Sinkhorn routing. In contrast, the non-decayed and decayed PBT$k$ MoE checkpoints go through a period of high routing imbalance immediately following the distribution shift. However, this period is short-lived: the PBT$k$ checkpoints recover to well-balanced MRI levels, better than those of SBT$k$, within 500 training steps. Subfigure (c) shows that this MRI spike can be mitigated with replay, though the benefit is negligible, as even the no-replay model recovers quickly. These results suggest that although SB is more robust to distribution shifts than PB, this robustness limits the MRI attainable. We hypothesize that the generally higher MRI of PB models may cause an uneven utilization of the MoE's parameters during training. Such a difference in expert utilization during training could possibly explain the differences in performance between PB and SB. Finally, the chaotic phase undergone by PBT$k$ checkpoints does not last long enough to forego the strong performance of these models.

## 5.2 Language modeling performance

Having established the benefits of replay and infinite learning rate schedules for continually pre-training MoEs, we now quantitatively verify the efficacy of these techniques by continually pre-training our MoEs on 200B tokens of Code and German Common Crawl data and evaluating their performance relative to two baselines. Specifically, we will compare the performance of the four continually pre-trained MoEs in our study to a FLOP-matched dense baseline and a fully re-trained PBT$k$ Granular MoE Baseline (the best-performing architecture in our study). Performance will be measured across 4 axes: validation loss on the pre-training and CPT datasets, English evaluation benchmarks (task 1), German and Code evaluation benchmarks (task

Table 1: **Aggregated benchmark results.** MoEs consistently outperform FLOP-matched dense baselines and exhibit less forgetting w.r.t. validation loss. Compared to the re-training baseline (blue), MoEs and the dense model match or exceed their performance. These results show MoEs adapt as well as dense models but forget less, likely due to their larger parameter count. All validation losses report log perplexity on a held-out validation set, forgetting (equivalent to backward transfer in Lopez-Paz & Ranzato (2017)) is calculated using validation loss, and downstream tasks report accuracy.

| Training Tokens | Model | Final Validation Loss (↓) | | | | | Downstream Evals. (↑) | | |
| | | FineWeb | Stack | German | Forgetting | AVG | English | German | Stack |
|---|---|---|---|---|---|---|---|---|---|
| 400B FineWeb | Dense Baseline | 2.881 | 4.028 | 3.741 | – | – | 49.84% | 23.54% | 0.00% |
| | SB Switch MoE | 2.711 | 3.861 | 3.495 | – | – | 54.14% | 23.11% | 0.00% |
| | PB Switch MoE | 2.699 | 3.872 | 3.451 | – | – | 54.45% | 23.37% | 0.00% |
| | SB Granular MoE | 2.664 | 3.690 | 3.404 | – | – | **55.71**% | 22.83% | 0.00% |
| | PB Granular MoE | 2.653 | 3.715 | 3.370 | – | – | 55.59% | 23.40% | 0.00% |
| 400B FineWeb → 200B Stack (30% Replay) | Dense Baseline | 2.939 | 1.026 | – | 0.059 | 1.982 | 49.21% | – | 7.37% |
| | SB Switch MoE | 2.757 | 0.944 | – | 0.046 | 1.850 | 51.76% | – | **9.09**% |
| | PB Switch MoE | 2.749 | 0.945 | – | 0.050 | 1.847 | 52.59% | – | 8.22% |
| | SB Granular MoE | 2.708 | **0.925** | – | **0.044** | 1.816 | 53.51% | – | 7.45% |
| | PB Granular MoE | 2.699 | **0.924** | – | 0.046 | 1.811 | 53.70% | – | 7.81% |
| 400B FineWeb ∪ 200B Stack | Dense Baseline | 2.866 | 1.050 | – | – | 1.958 | 49.57% | – | 3.76% |
| | PB Granular MoE | **2.630** | 0.935 | – | – | **1.782** | 54.79% | – | 7.44% |
| 400B FineWeb → 200B German (40% Replay) | Dense Baseline | 2.946 | – | 1.367 | 0.066 | 2.157 | 48.15% | 25.27% | – |
| | SB Switch MoE | 2.749 | – | 1.142 | 0.039 | 1.946 | 51.99% | 27.57% | – |
| | PB Switch MoE | 2.741 | – | 1.129 | 0.042 | 1.935 | 51.25% | 26.50% | – |
| | SB Granular MoE | 2.701 | – | 1.118 | **0.037** | 1.910 | 53.35% | **28.57**% | – |
| | PB Granular MoE | 2.690 | – | **1.099** | **0.037** | 1.895 | 53.61% | 27.65% | – |
| 400B FineWeb ∪ 200B German | Dense Baseline | 2.938 | – | 1.390 | – | 2.164 | 48.42% | 25.45% | – |
| | PB Granular MoE | 2.669 | – | 1.120 | – | **1.895** | 53.94% | 27.59% | – |

2), and MRI of final checkpoints. Note that the main conclusions of this section are succinctly summarized in Figure 1.

**Validation Loss.** Table 1 reports validation loss (e.g., log perplexity on a fixed held-out validation set) results for the main models in our study, while extended results are reported in Table 6 of the appendix. We observe that all MoEs outperform the FLOP-matched dense baseline during pre-training and CPT. Within the MoEs, we observe that PBT$k$ MoEs consistently outperform SBT$k$ MoEs and that Granular MoEs consistently outperform switch MoEs across pre-training and CPT. These findings are consistent with the literature on Granular MoEs (Ludziejewski et al., 2024; Dai et al., 2024), but we believe this is the first time that SBT$k$ routing has been shown to underperform PBT$k$ routing. Since the PBT$k$ Granular MoE achieves the best performance, we use it as our full re-training baseline. Compared to full re-training, our Granular CPT MoEs consistently have higher FineWeb validation loss but achieve lower downstream validation loss with a similar average validation loss. Similar results are found when comparing the CPT dense baseline to its full re-training counterpart. These results demonstrate that the continual learning abilities of MoEs w.r.t. validation loss are on par with dense models for adaptation and are slightly superior in terms of forgetting, likely due to their larger total parameter count.

**English Evaluation results.** Table 1 presents average accuracy, while Table 4 details per-benchmark results. We select benchmarks where models of our scale ($570M$ active parameters) achieve non-trivial accuracy to maximize signal. Each model is evaluated zero-shot on benchmarks covering Commonsense Reasoning, Reading Comprehension, Scientific Question Answering, and Math: HellaSwag (Zellers et al., 2019), Winogrande (Sakaguchi et al., 2019), PIQA (Bisk et al., 2019), ARC-Easy, ARC-Challenge (Clark et al., 2018), SWAG (Zellers et al., 2018), LAMBADA (OpenAI) (Storks et al., 2019), SciQ (Johannes Welbl, 2017), PubMedQA (Jin et al., 2019), and MathQA (Amini et al., 2019) (see Sec. D.1 for details). We find that models trained solely on FineWeb outperform all others, including full re-training baselines. Granular MoEs surpass switch MoEs. CPT models trained on Stack perform similarly to those trained on German. Compared to full re-training, CPT models achieve nearly identical results (within ∼ 1%). The dense baseline also matches its full re-training counterpart, indicating that MoEs have similar continual learning abilities on pre-training evaluations while benefiting from improved sample efficiency.

**German Evaluation results.** Table 1 shows average German evaluation performance, while table 5 of the appendix provides a per-benchmark breakdown. We use GPT-3–translated German versions of

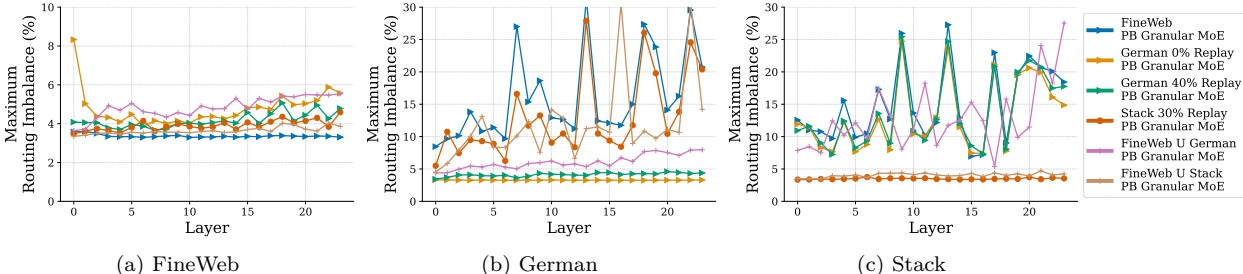

(a) FineWeb        (b) German        (c) Stack

Figure 4: **Layer-wise Maximum Routing Imbalance (MRI) for Granular MoEs.** We report MRI (eq. 1) on each dataset's 20M-token test set. MRI is consistently lower for Penalty-Balanced MoEs than Sinkhorn-Balanced MoEs. Continual pre-training on FineWeb incurs minimal MRI increase, even with 0% replay. MoEs are most unbalanced with out-of-distribution data (e.g., non-German models in (b) and non-code models in (c)).

HellaSwag, ARC-Challenge, and TruthfulQA, evaluating each zero-shot (Plüster, 2023). German-trained models outperform English-only ones, and German-trained MoEs surpass the FLOP-matched dense baseline. Among MoEs, modules using the same training tokens perform similarly. CPT MoEs and the full re-training baseline differ by $< 1\%$ accuracy, with no clear winner. The dense baseline also performs comparably to full re-training, demonstrating that the continual learning abilities of MoEs w.r.t. German evaluations are on par with dense models while benefiting from improved sample efficiency.

**Code Evaluation results.** Table 1 presents average Code evaluation performance, while Table 3 of the appendix provides a pass@k breakdown ($k \in \{1, 10, 50, 100, 150, 200\}$). Our models are evaluated on Python code-generation tasks from HumanEval (Chen et al., 2021), as Python is well-represented in our Stack CPT dataset (Table 10). English-trained models can not solve any problem, whereas stack-trained models achieve non-trivial accuracy. Unlike for other performance metrics, CPT switch MoEs slightly outperform their granular counterparts. Compared to full re-training, all CPT MoEs perform marginally better, while the CPT dense model exceeds its baseline by over 3%. We attribute this unexpected improvement to evaluation noise and training variance, given the models' similar validation loss. These results suggest MoEs match dense models in continual learning for code evaluation when accounting for MoEs' improved sample efficiency.

**Routing imbalance during and after continual pre-training.** Figure 4 shows the layer-wise Maximum Routing Imbalance (MRI) for Granular MoEs across FineWeb (a), German (b), and stack (c), while Figure 16 reports MRI for all MoEs. We include a 0% replay baseline for each MoE CPT on German to highlight replay's impact on MRI.

In subfigure (a), Penalty-Balanced MoEs consistently have lower MRI than Sinkhorn-Balanced MoEs across all architectures, and granular MoEs exhibit lower and more stable MRI within architectures. On FineWeb, continual pre-training causes only a slight MRI increase relative to the pre-trained checkpoint, even for the 0% replay model, except in its first layer. Interestingly, all German-trained MoEs show higher MRI on FineWeb than their Stack-trained counterparts, with the full re-training baseline, surprisingly, having the highest. This suggests there may be more routing interference between English and German datasets and that continual pre-training may help reduce MRI across distributions, possibly due to the use of *CosineInf* vs. *Cosine Annealing* schedules for CPT and re-training, respectively.

Granular MoEs also reduce routing imbalance on German (b) and Stack (c) datasets. MoEs become most unbalanced with out-of-distribution data (e.g., non-German models in (b) and non-code models in (c)). Similar trends hold for Switch MoEs, with an additional finding: high MRI is common in early layers of Switch MoEs, independent of training/testing distributions, unlike Granular MoEs. These results show that PBT$k$ and SBT$k$ MoEs are robust to distribution shifts w.r.t. MRI and can even outperform re-training baselines, suggesting that continually pre-training MoEs should have no negative impact on inference latency.

*In summary, we find that, across three measures of performance, MoEs continually pre-trained with replay and infinite LR schedules can match the performance of a full re-training baseline and, thus, they have similar CPT abilities to a FLOP-matched dense baseline without any inhibition from their routers. Moreover, we show that continually pre-training MoEs has no negative impact on MRI compared to re-training.*

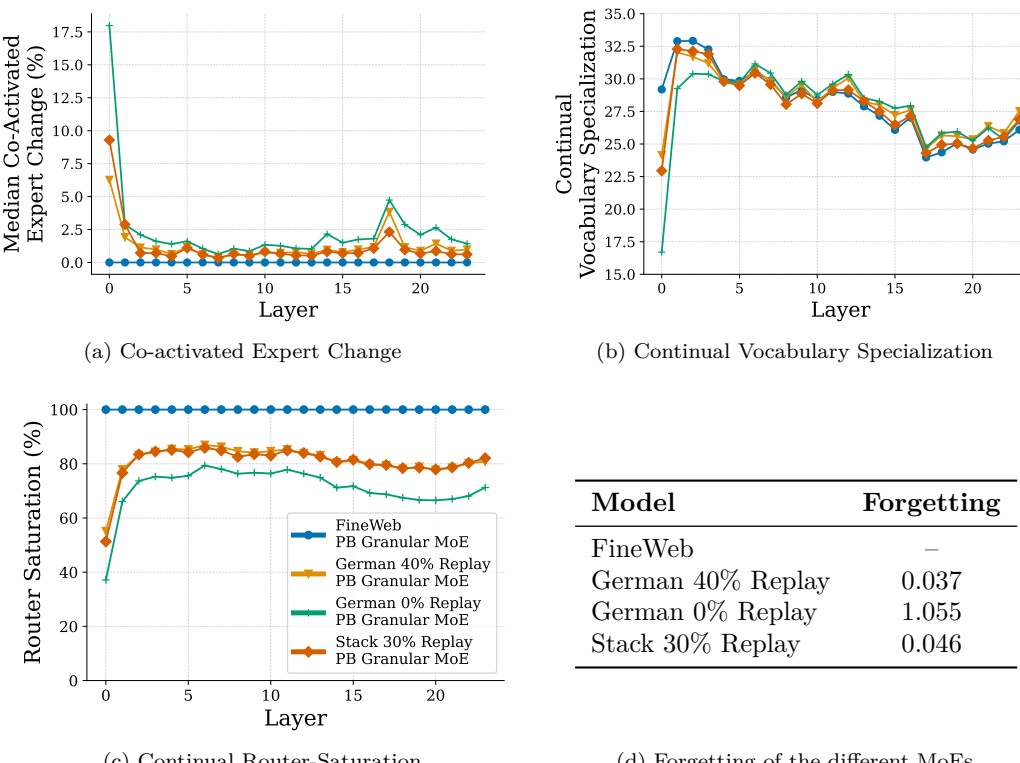

(a) Co-activated Expert Change

(b) Continual Vocabulary Specialization

(c) Continual Router-Saturation

(d) Forgetting of the different MoEs

| Model | Forgetting |
|---|---|
| FineWeb | – |
| German 40% Replay | 0.037 |
| German 0% Replay | 1.055 |
| Stack 30% Replay | 0.046 |

Figure 5: **Layer-wise analysis of routing changes during CPT.** Our goal is to understand how routing decisions change from the pre-trained checkpoints to final checkpoints after continual pre-training. To this end, we analyse changes in routing behaviour from 3 perspectives: which experts tend to be activated together (a), the tendency for certain vocabulary tokens to be routed to certain experts (b), and how close routing decisions of the pre-trained checkpoint are from CPT checkpoints (c). To provide context to these metrics, we remind the reader of the forgetting (e.g., from table 1) for each model shown in the plots. We observe that the no-replay baseline changes the most in early layers and forgets the most, suggesting that more drastic changes in initial layers may be linked to forgetting.

## 5.3 Analyzing changes in routing behaviour due to CPT

In the following section, we analyze changes in routing behaviour resulting from continual pre-training. Specifically, we record routing decisions of the MoE checkpoints before and after continual pre-training on $20,000,000$ tokens of held-out test data from FineWeb, Stack, and German. To understand how routing decisions change during CPT, we adapt three routing behavior metrics from Muennighoff et al. (2024) to the continual pre-training setting: Router Saturation, Vocabulary specialization, and Expert co-activation. We will provide brief descriptions of each in what follows, with formal definitions available in the appendix (Sections D.3.1, D.3.2, and D.3.3).

**Continual Router-Saturation** Router Saturation (RS) is the percentage of routing decisions at iteration $t$ that match those of the final checkpoint (Muennighoff et al., 2024). We extend this metric to multiple training phases for continual pre-training. Figure 5 (c) shows RS between the pre-training and CPT checkpoints for Stack and German Granular PBT$k$ MoEs. RS is lowest in early layers, peaks at layers $2-13$, and slightly drops after layer 13. The 0% replay German checkpoint has RS consistently $10-15\%$ lower than the 40% replay checkpoint across all layers. Note that despite adapting well to German, only the no-replay checkpoint suffers significant forgetting on FineWeb. These results suggest that CPT adaptation is most pronounced in layers $0-2$ and $13-23$ and that forgetting has the same pattern but is correlated with lower overall router saturation.

**Continual Vocabulary Specialization** Vocabulary Specialization (VS) quantifies how often a token from a dataset is routed to a specific MoE expert relative to its total occurrences (Muennighoff et al., 2024).

By assigning each token in the model's vocabulary to the expert that processes it the most frequently, we can create a *one-to-many mapping* between experts and vocabulary entries for an MoE layer. Then, we can calculate the average VS of each expert by averaging its vocabulary specialization across its assigned tokens. Vocabulary specialization within a layer is calculated by taking the mean average VS across experts in that layer. To compare specialization across model checkpoints, we can re-use the *one-to-many mapping* of a previous checkpoint and measure how the specialization w.r.t. this mapping has changed during CPT. Figure 5 shows VS w.r.t. pre-training checkpoints using FineWeb data. VS is notably lower in layers 0-4 after CPT, while there is almost no discernible change in VS for layers $5 - 23$. The zero-replay checkpoint exhibits the lowest VS in layers 0-4, correlating with its weaker FineWeb performance and suggesting that excessive VS shifts in early layers may contribute to forgetting.

**Co-activated Expert Change** Expert co-activation between two experts $E_i$ and $E_j$ is defined as the ratio of times they are activated together to the total activations of $E_i$ over a dataset (Muennighoff et al., 2024). This metric applies only to MoEs with $k \geq 2$ active experts. A co-activation matrix can be constructed for each ordered expert pair in a layer. To compare expert co-activation across model checkpoints, we compute co-activation matrices for all layers of two checkpoints $(C^{(1)}, C^{(2)})$ and measure absolute changes by computing statistics of the entries in their element-wise absolute differences matrix ($|C^{(1)} - C^{(2)}|$). Figure 5 (a) shows the median co-activation change between the pre-training and CPT checkpoints. Early layers (0-1) exhibit the largest changes, with a consistent spike at layer 18 for all CPT models and slightly higher median changes in layers $13 - 23$. Among CPT checkpoints, the no-replay variant shows the most significant co-activation shifts. Despite all checkpoints adapting well to new distributions, only the no-replay checkpoint experiences substantial forgetting on FineWeb. These findings suggest that adaptation during CPT correlates with co-activation changes in early $(0 - 2)$ and later $(13 - 23)$ MoE layers and that more pronounced changes correlate with higher forgetting.

*In summary, results across all three metrics reveal that routing decisions change most in the early layers of Granular MoE transformers, while changes in other MoE layers are observed for expert co-activation and router saturation but not for Vocabulary specialization. Of all models, the no-replay baseline changes the most in early layers and forgets most, suggesting that more drastic changes in initial layers may be linked to forgetting.*

## 6    Conclusion

We have conducted a comprehensive empirical study on the continual pre-training of decoder-only MoE transformer language models. Our large-scale experiments, involving 2B parameter MoEs trained on 600B tokens, demonstrate that both Penalty-Balanced (PBT$k$) and Sinkhorn-Balanced (SBT$k$) routing algorithms exhibit surprising system-level resilience to distribution shifts, maintaining balanced loads as measured by the novel Maximum Routing Imbalance metric. We established that MoEs preserve their sample efficiency advantage over FLOP-matched dense models during CPT and that, when using infinite LR schedules and replay, a Granular PBT$k$ MoEs can match the performance of fully re-trained baselines across German and Code transitions, at a fraction of the computational cost. Finally, we saw that early MoE layers change the most during CPT, suggesting that future work could investigate special treatment of these layers for improved performance. Collectively, our findings establish MoEs as strong continual learners for text, comparable to dense models, and underscore their potential as scalable, adaptable foundation models for language.

## Acknowledgments

We acknowledge support from NSERC Discovery Grant RGPIN- 2021-04104 [E.B.], FRQNT New Scholar [*E.B.*], the Canada CIFAR AI Chair Program [I.R.], the Canada Excellence Research Chairs Program [I.R.], and the FRQNT Doctoral (B2X) scholarship [B.T.]. This research was enabled in part by compute resources provided by Mila (mila.quebec). We would also like to thank the GenAI support team at Capital One for their assistance and, in particular, Dhantha Gunarathna. We also thank Akshaj Kumar Veldanda, Andrei Mircea, Hanyang Zhao, and Supriyo Chakraborty for helpful discussions throughout.

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

## Contents

# A  Extended Background

The following section is complementary to section 2 of the main manuscript, providing additional background for our paper.

## A.1  Continual pre-training of LLMs

Continual pre-training extends pre-training to multiple new distributions. Concretely, continual pre-training occurs when a models is trained on a sequence of datasets $\mathcal{D}_0, \mathcal{D}_1, \ldots, \mathcal{D}_N$ with different distributions, $N \geq 2$, and each dataset is sufficiently large (e.g., $> 100$B tokens in the case of language) (Ibrahim et al., 2024). Note that the large data scale, here, distinguishes this setting from supervised fine-tuning or instruction tuning where the amount of data is much smaller. Typical application settings of continual pre-training are adapting existing pre-trained models on newly available data or enhancing their capabilities in a specific domain. We will now discuss well-established techniques for continually pre-training dense transformers.

**LR Re-warming and Re-decaying.** Many open-source LLMs follow a linear warmup and cosine annealing schedule during pre-training, which reaches a large maximum learning rate, $\eta_{max}$, early on in training and subsequently decays the learning rate to a small minimum value, $\eta_{min}$ (Hoffmann et al., 2022; Loshchilov & Hutter, 2017; Rae et al., 2021). Naively continuing training at $\eta_{min}$ or $\eta_{max}$ either leads to too little adaptation or too much forgetting. Instead, Ibrahim et al. (2024) show that Re-warming and Re-decaying the learning rate during CPT is critical for strong continual learning performance.

**Infinite LR schedules.** While Re-warming and Re-decaying the learning rate following a cosine decay schedule was found to be a good solution when starting from a fully decayed checkpoint, Ibrahim et al. (2024) remark that this strategy incurs forgetting, due to the large LR increase, even when continually pre-training on the same distribution. To circumvent this, the authors propose infinite learning rate schedules that allow for a smooth transition in learning rate between continual learning phases and are not bound to a fixed number of training steps. These techniques were subsequently confirmed to work well in settings with multiple distribution shifts (Janson et al., 2025).

**Replay.** Replaying previous data has been a longstanding tool for mitigating catastrophic forgetting (Wang et al., 2024b). In our experiments, we replay previously seen data for this purpose, designating any model using the technique with the suffix "$X\%$ Replay". Here, $X$ represents the percentage of samples in a given batch that were replayed from the previous distribution. To match compute across different replay budgets, we do not increase the token budget when increasing the amount of replay. Instead, we decrease the amount of new data seen during CPT.

## A.2  Mixture of experts transformer language models

Sparsely-activated MoE transformers differ from their dense counterparts by dynamically routing tokens in a sequence, $X \in \mathbb{R}^{S \times H}$, to different experts $\{\mathrm{FFN}_{i,j}(\cdot)\}_{i=0}^N$ as opposed to a single FFN. Here $S$ is the sequence length, $H$ is the transformer's hidden dimension, $j$ indexes over the transformer's blocks, and $N$ is the number of experts per block. This is often referred to as an MoE layer (Shazeer et al., 2017). Typically, these layers are used in place of Feed Forward Networks (FFN) in each transformer block (Fedus et al., 2022; Dai et al., 2024), however, recent works (Shen et al., 2024; Zhang et al., 2022) have also replaced the query and output matrices of multi-head self-attention layers with MoE layers. In what follows, we exclusively study MoE transformers that replace FFNs at each block with MoE layers, similarly to state-of-the-art recent work (Dai et al., 2024; Team, 2024; Muennighoff et al., 2024; DeepSeek-AI et al., 2025b;a). Moreover, we also study the recent trend of using more granular experts and shared experts (Dai et al., 2024; Team, 2024; Muennighoff et al., 2024; He, 2024; Liu et al., 2023b; Ludziejewski et al., 2024; Rajbhandari et al., 2022; DeepSeek-AI et al., 2025b;a).

Algorithms for dynamically selecting among experts, known as routing algorithms (Roller et al., 2021; Shazeer et al., 2017; Zoph et al., 2022; Clark et al., 2022; Lewis et al., 2021), are central to MoEs. A key consideration

for token-choice (we do not consider expert-choice as it is incompatible with autoregressive generation) routing algorithms is achieving a balanced load across experts in a given layer. Without enforcing a balanced load, the router may collapse to only choosing a single or a few experts, leading to poor parameter utilization and higher latency proportional to the load of the most burdened expert (Zhou et al., 2022).

In this work, we focus on studying two prominent Top-$k$ routing algorithms from recent state-of-the-art works, which we refer to as Penalty-Balanced Top-$k$ (PBT$k$) routing (Shazeer et al., 2017; Dai et al., 2024; Zoph et al., 2022; Fedus et al., 2022) and Sinkhorn-Balanced Top-$k$ (SBT$k$) routing (Clark et al., 2022; Anthony et al., 2024). Both algorithms define the router $R(x) = Wx : \mathbb{R}^H \to \mathbb{R}^e$ to be a simple linear projection to the space of experts. Expert probabilities are computed by applying a softmax to the router's output: $p(x) = \texttt{softmax}(SB(R(x)))$. Where $SB(\cdot)$ is the Sinkhorn load balancing function in the case of SBT$k$ routing or the identity otherwise.

Ranked by $p(x)$, the top-$k$ experts are selected for each token, where $k$ is a hyperparameter selected before training. For a single token, the output of MoE layer $j$, $f_{\text{MoE}_j}$ is computed as follows:

$$f_{\text{MoE}_j}(x) = \text{SFFN}_j(x) + \frac{\sum_{i \in I_k(x)} p_i(x) \cdot \text{FFN}_{i,j}(x)}{\sum_{i \in I_k(x)} p_i(x)}.$$

Where $I_k(x) = \{i \mid p_i(x) \in \text{Top k elements of } p(x)\}$ and SFFN is a shared expert (Rajbhandari et al., 2022; Dai et al., 2024) if one is used or the identity otherwise. While they both route tokens to the top-$k$ experts, the PBT$k$ and SBT$k$ routing algorithms differ in how they balance the load across experts.

**Penalty-Balanced Top-$k$ Routing.** PBT$k$ methods in the literature (Shazeer et al., 2017; Fedus et al., 2022; Zoph et al., 2022; Dai et al., 2024) add penalty terms to the overall loss to encourage a balanced load across experts. The *auxiliary loss* has become the most popular such penalty and is used in conjunction with the *z-loss* in several recent state-of-the-art MoEs (Dai et al., 2024; Team, 2024). Briefly, the auxiliary loss is minimized when the router assigns an equal proportion of tokens in a given batch to each expert in a given block, while the z-loss penalizes large-magnitude router logits. The latter has been shown to promote numerical stability in larger models (Zoph et al., 2022). Given their combination in recent SOTA MoE LLMs (Zoph et al., 2022; Dai et al., 2024), we exclusively study MoEs that combine both *auxiliary loss* and *z-loss*, referring to them as PBT$k$ MoEs.

**Sinkhorn-Balanced Top-$k$ Routing.** SBT$k$ routing casts the assignment of tokens to experts as a linear assignment problem which corresponds to a well-studied problem in optimal transport, namely "the regularized Kantorovich problem of optimal transport" (Clark et al., 2022). The Sinkhorn-knopp algorithm (Knopp & Sinkhorn, 1967) provides an approximate solution to this problem, which can be efficiently computed on GPUs. In practice, this corresponds to adjusting routing probabilities (e.g, according to $SB(\cdot)$, see (Clark et al., 2022) section B.2.1 for details) such that a relatively balanced load is obtained without deviating *too much* from greedy Top-$k$ routing.

# B Extended related work

The following section is complementary to section 3 of the main manuscript, providing a more comprehensive summary of the related work.

## B.1 Mixture of experts language models

Mixture of experts language models have a long history with fundamental ideas dating back several decades (Collobert et al., 2003; Jacobs et al., 1991). More recently, in the context of large-scale language modeling, the mixture-of-experts layer (Shazeer et al., 2017) was introduced to substantially increase the capacity of an LSTM language model with little detriment to efficiency. The authors also introduced a load-balancing penalty to encourage even utilization of experts. Subsequently, Fedus et al. (2022) refined the penalty, renamed *auxiliary loss*, which has become a central component of modern MoEs. Follow-up works have focused on massively scaling up MoE LLMs, improving the routing algorithms of these models, improving the quality of the router's gradient estimate, and making architectural improvements to these models. Lepikhin et al. (2021) introduce the MoE layer into the transformer architecture, using two activated

experts (thought to be necessary for nontrivial router gradients) and scaling to an unprecedented 600B parameter scale. Subsequently, Fedus et al. (2022) introduced the Switch Transformer, showing that it is possible to scale MoEs beyond 1T parameters despite training them with only a single active expert.

Other works have focused on developing novel routing algorithms. Lewis et al. (2021) cast routing as a linear assignment problem and leverage Hungarian matching in their routing algorithm. Clark et al. (2022) use Sinkhorn's algorithm to approximately solve the assignment problem on GPUs, resulting in a faster algorithm. Anthony et al. (2024) introduce a favorable initial condition to improve the convergence of the iterative Sinkhorn solver, further reducing the cost of Sinkhorn routing. Roller et al. (2021) introduces a deterministic routing algorithm based on hash layers. Zoph et al. (2022) introduces a loss penalty to promote stability in large-scale MoE routing. Wang et al. (2024a) introduces the first learned routing mechanism that uses neither an entropy regularizer nor an assignment-based approach to balance expert utilization in token-choice routing. Zhou et al. (2022) introduce Expert Choice Routing, a routing paradigm where each expert receives a balanced load and the routing algorithm decides which tokens to send to each of the experts; while it obtains strong performance and automatically achieves a balanced load, ECR is incompatible with autoregressive generation so we don't consider it in this work. Other works propose methods to better approximate the full router gradient (Panda et al., 2024; Liu et al., 2024; 2023a).

Finally, a recent trend of using MoE experts with finer-grained intermediate sizes has shown notable performance gains when compared to using the full intermediate FFN size, as was originally done (Shazeer et al., 2017; Fedus et al., 2022; Lepikhin et al., 2021). Liu et al. (2023b) first observed that utilizing smaller expert layers improves perplexity. Subsequently, researchers have explored scaling laws for fine-grained MoEs at small scale (Ludziejewski et al., 2024), pre-trained and released SOTA MoEs that leverage the fine-grained expert architecture (Dai et al., 2024; Team, 2024; Muennighoff et al., 2024), and pushed the idea of thinner experts to its limit, exploring MoEs with millions of experts (He, 2024). While we have reviewed the most relevant works to ours, there are many more works that we have not had the chance to mention here. We refer the avid reader to a recent and comprehensive survey of the area (Cai et al., 2024).

## B.2 Continual pre-training of dense foundation models

Continual pre-training of foundation models has the same objectives as continual learning (French, 1999), except that it is applied to large-scale pre-training tasks, which are mainly self-supervised. Several existing works study continual learning in settings relevant to continual pre-training. They find that self-supervised pre-training benefits from reduced forgetting (Cossu et al., 2022; Davari et al., 2022), that pre-trained models forget less than their randomly initialized counterparts (Mehta et al., 2023), that forgetting improves as model scale is increased (Ramasesh et al., 2022), and that wider models tend to forget less than deeper models (Mirzadeh et al., 2022). In the context of LLM fine-tuning, (Scialom et al., 2022) shows that little replay is needed to prevent forgetting when fine-tuning on small amounts of instruction-tuning data. In the context of large-scale (with respect to data) continual pre-training for LLMs, Gupta et al. (2023) highlights the importance of rewarming the learning rate when models are pre-trained from a checkpoint that has been decayed to a small learning rate. Following up on their work, Ibrahim et al. (2024) establish the effectiveness of learning rate re-warming, LR re-decaying, and replay for large-scale continual pre-training of LLMs. Concurrently, Garg et al. (2023) establishes the performance of the same techniques for CLIP models. Shortly thereafter, Parmar et al. (2024) scale continual pre-training for dense decoder-only transformers further, showing that a 15B parameter model pre-trained for 8T tokens can be effectively pre-trained on 1T tokens of incoming data.

## B.3 Continual pre-training of MoE LLMs.

To the best of our knowledge, only a single work exists exploring the large-scale continual pre-training of MoEs LLMs, while the majority of the literature focuses on upcycling or growing MoEs for continual pre-training.

In a concurrent pre-print DeepSeek-CoderV2 (DeepSeek-AI et al., 2024), shows that they can continue from a checkpoint the training of a MoE LLM. However, this is only shown for one instance and the analysis of the MoE routing behavior is not discussed. Furthermore, there is no comparison to a FLOP-matched dense

model, making it challenging to assess whether the sample efficiency of MoE LLMs is maintained during continual pre-training.

Continual pre-training methods for MoEs that are less related to our work generally focus on fine-tuning MoE LLMs on small amounts of data (Wang et al., 2024c) or growing MoEs (Komatsuzaki et al., 2023; Zhu et al., 2024; Sukhbaatar et al., 2024; Gritsch et al., 2024; Chen et al., 2023). Wang et al. (2024c) study MoE-specific techniques for parameter-efficient fine-tuning (PEFT). Zhu et al. (2024) proposes a technique to create an MoE by splitting the FFNs of an existing dense transformer and subsequently continually pre-training it. Sukhbaatar et al. (2024) proposes to continually pre-train a dense LLM on multiple different datasets, gather the FFN layers from different continually pre-trained models to form MoE layers, merge the parameter tensors other than FFN layers, and subsequently continually pre-train the merged model to learn routing in the MoE part. Gritsch et al. (2024) propose a similar method to train new expert layers that uses domain embeddings from a pre-trained embedding model as the identifier for a domain's experts, allowing the domain embeddings to provide an inductive bias that can help with adding new experts. While these methods allow for improving the capabilities of MoEs with new data, they focus on first upcycling dense models, whereas we focus on updating MoEs pre-trained from scratch.

## C  Training timings

In the interest of completeness, we provide training timings for each model architecture in our study. We would like to preface this section with the following disclaimer: all the step times that we report in our study are specific to our code and the libraries that we use, but are not reflective of the best performance achievable. Timings are subject to vary based on model size, interconnect speed, training precision, accelerator used, implementation, etc. With this in mind, in table 2, we report the mean and standard deviation timing in milliseconds (ms) of different operations across 1000 training steps on 64 A100 GPUs. Specifically, we time the forward pass, backward pass, optimizer step, and dataloading time. An aggregate time is reported in the last column. We also report the MFU achieved in TFLOPs and report the throughput in samples per second. All our experiments use code from the GPT-NeoX library (Andonian et al., 2023) and leverage the megablox grouped GEMM kernel[1] (Gale et al., 2023). We observe that, in our specific implementation, all MoEs take approximately twice as long per step as the dense model, that granular MoEs have slower forward and backward times compared to Switch MoEs, and that Sinkhorn-balanced MoEs have slower forward and backward times compared to Penalty-balanced MoEs. This reveals some non-negligible downsides (e.g. as reflected in step times) to training MoEs compared to a FLOP-matched dense model, including: higher memory, longer optimizer step, and higher communication costs. Despite these drawbacks, Du et al. (2024) find that the performance benefits of MoEs still exceed those of dense models, even when accounting for the additional overhead when training MoEs.

Table 2: **Time of different operations during pre-training for each model architecture in our study.** We report the mean and standard deviation timing in milliseconds (ms) of different operations across 1000 training steps on 64 A100 GPUs. We use a global batch size of 1024 and a sequence length of 2048. Specifically, we time the forward pass, backward pass, optimizer step, and dataloading time. An aggregate time is reported in the last column. We also report the MFU achieved in TFLOPs and report the throughput in samples per second. We observe that, in our specific implementation, all MoEs take approximately twice as long per step as the dense model, that granular MoEs have slower forward and backward times compared to Switch MoEs, and that Sinkhorn-balanced MoEs have slower forward and backward times compared to Penalty-balanced MoEs.

| Model | Forward | Backward | Optimizer | Data | Samples/Sec. | MFU(TFLOPs) | Total Time(ms) |
|---|---|---|---|---|---|---|---|
| Dense Baseline | $318.02 \pm 1.86$ | $518.60 \pm 17.73$ | $39.48 \pm 5.55$ | $3.72 \pm 0.44$ | $1156.88 \pm 52.63$ | $111.32 \pm 5.06$ | 879.83 |
| PB Switch MoE | $449.36 \pm 14.97$ | $963.53 \pm 4.06$ | $101.66 \pm 0.69$ | $2.42 \pm 0.33$ | $671.93 \pm 24.63$ | $86.21 \pm 3.16$ | 1516.97 |
| SB Switch MoE | $494.28 \pm 11.35$ | $1012.38 \pm 6.87$ | $101.58 \pm 0.70$ | $2.59 \pm 1.36$ | $631.14 \pm 33.90$ | $80.97 \pm 4.35$ | 1610.83 |
| PB Granular MoE | $485.05 \pm 13.66$ | $1091.38 \pm 3.68$ | $100.66 \pm 3.08$ | $2.42 \pm 0.33$ | $606.85 \pm 21.53$ | $77.86 \pm 2.76$ | 1679.50 |
| SB Granular MoE | $541.62 \pm 7.14$ | $1144.85 \pm 5.72$ | $100.27 \pm 1.47$ | $2.37 \pm 0.33$ | $569.72 \pm 19.27$ | $73.09 \pm 2.47$ | 1789.10 |

---

[1] https://github.com/tgale96/grouped_gemm

# D Extended experimental results

In the following section, we provide extended experimental results from the paper in a non-summarized format to enhance the reproducibility of our manuscript and allow the reader to dive into whichever details may most interest them.

## D.1 Language model evaluation benchmarks

We evaluate the language models in our study on English, Code, and German evaluation tasks. Please note that our goal is not to achieve SOTA performance on these benchmarks; none of our models have been aligned or fine-tuned to improve performance. Instead, we seek to evaluate their performance within the context of our controlled scientific study. Given the scale of our language models (at most 570M active parameters), we carefully select evaluation tasks that show non-trivial evaluation results; that is, we choose tasks for which the models in our suite achieve above random chance accuracy.

Selected English evaluation tasks:

- **Commonsense Reasoning (0-shot):** HellaSwag (Zellers et al., 2019), Winogrande (Sakaguchi et al., 2019), PIQA (Bisk et al., 2019), ARC-Easy, ARC-Challenge (Clark et al., 2018), SWAG (Zellers et al., 2018)
- **Reading Comprehension (0-shot):** LAMBADA (OpenAI) (Storks et al., 2019)
- **Scientific Question Answering (0-shot):** SciQ (Johannes Welbl, 2017), PubMedQA (Jin et al., 2019)
- **Math (0-shot):** MathQA (Amini et al., 2019)

Selected German evaluation tasks translated from the corresponding English language tasks using the GPT 3.5 API (Plüster, 2023).

- **Commonsense Reasoning (0-shot):** HellaSwag-DE (Zellers et al., 2019), ARC-Challenge-DE (Clark et al., 2018)
- **Reading Comprehension (0-shot):** TruthfulQA-DE (Joshi et al., 2017)

Code evaluation tasks

- **Python:** Human Eval (pass@1-200)

Tables 4, 3, and 5 report the performance of models in our study on English, Code, and German evaluation benchmarks.

Table 3: **Human Eval after pre-training on FineWeb and continual pre-training on Stack**. We report the percentage of problems for which at least one generated solution passes all tests. We observe that all English-only models generate only incorrect solutions, while the models continually pre-trained on code and the full re-training baselines achieve non-trivial accuracy. Interestingly, the SB Switch MoE performs best of all across all pass thresholds. However, given the generally poor performance of the models overall, we attribute differences within a dataset type to random chance.

| Training Tokens | Model | pass@1 | pass@10 | pass@50 | pass@100 | pass@150 | pass@200 | Mean |
|---|---|---|---|---|---|---|---|---|
| | Dense Baseline | 0.00% | 0.00% | 0.00% | 0.00% | 0.00% | 0.00% | 0.00% |
| | SB Switch MoE | 0.00% | 0.00% | 0.00% | 0.00% | 0.00% | 0.00% | 0.00% |
| 400B FineWeb | PB Switch MoE | 0.00% | 0.00% | 0.00% | 0.00% | 0.00% | 0.00% | 0.00% |
| | SB Granular MoE | 0.00% | 0.00% | 0.00% | 0.00% | 0.00% | 0.00% | 0.00% |
| | PB Granular MoE | 0.00% | 0.00% | 0.00% | 0.00% | 0.00% | 0.00% | 0.00% |
| | Dense Baseline | 0.21% | 1.94% | 6.87% | 9.96% | 11.85% | 13.41% | 7.37% |
| | SB Switch MoE | **0.24%** | **2.19%** | **8.06%** | **12.15%** | **14.85%** | **17.07%** | **9.09%** |
| 400B FineWeb → 200B Stack | PB Switch MoE | 0.21% | 1.93% | 7.28% | 11.10% | 13.56% | 15.24% | 8.22% |
| | SB Granular MoE | 0.18% | 1.69% | 6.50% | 10.01% | 12.30% | 14.02% | 7.45% |
| | PB Granular MoE | 0.16% | 1.51% | 6.30% | 10.40% | 13.24% | 15.24% | 7.81% |
| 400B FineWeb ∪ 200B Stack | Dense Baseline | 0.14% | 1.20% | 3.57% | 4.99% | 5.98% | 6.71% | 3.76% |
| | PB Granular MoE | 0.20% | 1.82% | 6.84% | 10.21% | 12.13% | 13.41% | 7.44% |

Table 4: **Languge models evaluation benchmarks after pre-training (English Web Data), continual pre-training (Code & German Web Data), and full re-training**. We report accuracy for all selected benchmarks. We observe that all MoEs and the dense baselines maintain similar relative performance before and after the distribution shift, showing that MoE LLMs' continual pre-training dynamics are similar to dense models with respect to forgetting on evaluation tasks. When comparing the mean evaluation performance of PB granular MoE to the full re-training baseline, we observe that the final performance is matched or very nearly matched with a substantially smaller computational cost.

| Training Tokens | Model | ARC-C | ARC-E | HellaSwag | LAMBADA OAI | MathQA | PIQA | PubMedQA | SciQ | SWAG | WinoGrande | Mean |
|---|---|---|---|---|---|---|---|---|---|---|---|---|
| | Dense Baseline | 23.55% | 54.25% | 39.99% | 47.55% | 23.52% | 71.98% | 51.80% | 82.70% | 47.59% | 55.49% | 49.84% |
| | SB Switch MoE | 26.79% | 60.48% | 45.60% | 53.44% | 24.52% | 74.16% | **62.00%** | 84.80% | 50.31% | 59.27% | 54.14% |
| 400B FineWeb (Annealed) | PB Switch MoE | **28.75%** | 61.15% | 46.16% | 54.22% | **25.86%** | 74.37% | 58.20% | 87.20% | 50.67% | 57.93% | 54.45% |
| | SB Granular MoE | 26.19% | **65.19%** | 48.10% | 56.24% | 24.66% | 74.92% | 61.10% | 89.30% | 51.35% | 60.06% | **55.71%** |
| | PB Granular MoE | **28.75%** | 62.92% | **48.45%** | 56.08% | 24.62% | 75.24% | 60.00% | 88.90% | **51.36%** | 59.59% | **55.59%** |
| | Dense Baseline | 22.35% | 52.02% | 39.12% | 49.04% | 23.15% | 70.46% | 52.50% | 81.90% | 47.09% | 54.14% | 49.18% |
| | SB Switch MoE | 24.23% | 57.91% | 44.26% | 51.72% | 24.52% | 73.12% | **60.30%** | 83.50% | 49.32% | 56.67% | 52.55% |
| 400B FineWeb (Non-Annealed) | PB Switch MoE | 26.54% | 60.86% | 44.59% | 53.21% | 23.99% | 73.39% | 52.30% | 85.80% | 49.98% | 56.27% | 52.69% |
| | SB Granular MoE | 26.54% | 62.63% | 46.85% | 55.87% | 24.56% | 73.67% | 58.60% | 87.70% | 50.72% | 59.04% | 54.62% |
| | PB Granular MoE | 27.82% | 61.20% | 46.52% | 55.46% | 24.36% | 74.97% | 58.80% | 86.80% | 50.87% | 58.64% | 54.54% |
| | Dense Baseline | 22.87% | 51.43% | 36.97% | 46.75% | 23.75% | 70.18% | 48.80% | 80.70% | 45.87% | 54.22% | 48.15% |
| | SB Switch MoE | 24.66% | 56.90% | 42.99% | 52.94% | 24.22% | 73.07% | 57.40% | 83.50% | 48.85% | 55.41% | 51.99% |
| 400B FineWeb → 200B German (40% Replay) | PB Switch MoE | 25.34% | 56.94% | 42.59% | 53.43% | 25.06% | 73.23% | 49.40% | 84.20% | 48.76% | 53.51% | 51.25% |
| | SB Granular MoE | 25.43% | 60.02% | 44.67% | 55.23% | 25.13% | 73.01% | 55.10% | 84.60% | 49.89% | **60.46%** | 53.35% |
| | PB Granular MoE | 27.05% | 60.52% | 44.66% | 54.43% | 24.56% | 73.88% | 58.40% | 85.70% | 49.70% | 57.22% | 53.61% |
| 400B FineWeb ∪ 200B German CC | Dense Baseline | 23.29% | 51.35% | 36.77% | 46.17% | 24.19% | 70.08% | 54.00% | 80.30% | 45.51% | 52.57% | 48.42% |
| | PB Granular MoE | 27.99% | 60.06% | 45.06% | 55.23% | 25.16% | 73.50% | 56.20% | 86.20% | 49.88% | 60.14% | 53.94% |
| | Dense Baseline | 22.01% | 52.95% | 37.49% | 46.98% | 22.91% | 71.06% | 55.40% | 83.70% | 45.76% | 53.83% | 49.21% |
| | SB Switch MoE | 22.87% | 55.98% | 42.51% | 52.84% | 24.12% | 72.80% | 55.80% | 85.80% | 48.78% | 56.83% | 51.76% |
| 400B FineWeb → 200B Stack (30% Replay) | PB Switch MoE | 26.28% | 59.01% | 42.78% | 53.17% | 24.32% | 73.50% | 55.10% | 86.20% | 49.07% | 56.43% | 52.59% |
| | SB Granular MoE | 26.54% | 60.19% | 44.57% | 55.44% | 24.39% | 73.01% | 55.60% | 85.90% | 49.83% | 59.59% | 53.51% |
| | PB Granular MoE | 25.43% | 60.27% | 44.88% | 54.94% | 25.36% | 73.78% | 56.90% | 88.00% | 49.62% | 57.85% | 53.70% |
| 400B FineWeb ∪ 200B Stack | Dense Baseline | 22.18% | 52.78% | 38.68% | 48.50% | 24.49% | 71.00% | 51.70% | 83.40% | 47.01% | 55.96% | 49.57% |
| | PB Granular MoE | 27.82% | 62.67% | 46.43% | **56.39%** | 25.66% | **75.35%** | 56.60% | **89.40%** | 50.85% | 56.75% | 54.79% |

Table 5: **German Language models evaluation benchmarks after pre-training (English Web Data), continual pre-training (Code & German Web Data), and full re-training**. We report accuracy for all selected benchmarks. We observe that all MoEs and the dense baseline improve performance on German after continual pre-training. When comparing to the full re-training baselines, we observe that the average performance of our continually pre-trained models are on par.

| Training Tokens | Model | Arc-C DE | Hellaswag DE | TruthfulQA DE (MC1) | Mean |
|---|---|---|---|---|---|
| 400B FineWeb | Dense Baseline | 18.52% | 26.78% | 25.34% | 23.54% |
| | SB Switch MoE | 18.60% | 26.87% | 23.87% | 23.11% |
| | PB Switch MoE | 18.94% | 26.56% | 24.60% | 23.37% |
| | SB Granular MoE | 18.34% | 26.78% | 23.38% | 22.83% |
| | PB Granular MoE | 18.43% | 27.05% | 24.72% | 23.40% |
| 400B FineWeb → 200B German CC | Dense Baseline | 19.28% | 32.53% | 23.99% | 25.27% |
| | SB Switch MoE | 21.76% | 35.74% | 25.21% | 27.57% |
| | PB Switch MoE | 20.73% | 35.77% | 23.01% | 26.50% |
| | SB Granular MoE | 22.61% | 36.90% | **26.19**% | **28.57**% |
| | PB Granular MoE | 22.70% | **37.23**% | 23.01% | 27.65% |
| 400B FineWeb ∪ 200B German CC | Dense Baseline | 20.05% | 31.45% | 24.85% | 25.45% |
| | PB Granular MoE | 21.33% | 35.74% | 25.70% | 27.59% |

## D.2   Training and validation loss

In the following sections, we present validation loss curves during and after pre-training and continual pre-training for all models in our study. Specifically, we report final validation loss in Table 6 and validation curves during training across figures 6, 7, and 8.

Table 6: **Final validation loss of MoEs and dense model after pre-training (English Web Data) and continual pre-training (Code & German Web Data)**. As expected, we observe that all MoE transformers outperform the dense baseline during pre-training and continual pertaining with respect to validation loss. Moreover, we observe that MoEs forget marginally less than their dense counterparts. Together, these results show the continual learning abilities of MoEs are on par with dense models in terms of adaptation and are slightly superior in terms of forgetting, possibly due to their larger total parameter count.

| Training Tokens | Model | Final Validation Loss | | | | |
| | | FineWeb | Stack | German | Forgetting | AVG |
|---|---|---|---|---|---|---|
| | Dense Baseline | 2.881 | 4.028 | 3.741 | – | – |
| | SB Switch MoE | 2.711 | 3.861 | 3.495 | – | – |
| 400B FineWeb (non-annealed) | PB Switch MoE | 2.699 | 3.872 | 3.451 | – | – |
| | SB Granular MoE | 2.664 | 3.690 | 3.404 | – | – |
| | PB Granular MoE | 2.653 | 3.715 | 3.370 | – | – |
| | Dense Baseline | 2.825 | 4.028 | 3.741 | – | – |
| | SB Switch MoE | 2.640 | 3.861 | 3.495 | – | – |
| 400B FineWeb (annealed) | PB Switch MoE | 2.628 | 3.872 | 3.451 | – | – |
| | SB Granular MoE | 2.595 | 3.690 | 3.404 | – | – |
| | PB Granular MoE | 2.582 | 3.715 | 3.370 | – | – |
| | Dense Baseline | 2.939 | 1.026 | – | 0.059 | 1.982 |
| | SB Switch MoE | 2.757 | 0.944 | – | 0.046 | 1.850 |
| 400B FineWeb → 200B Stack 30% Replay | PB Switch MoE | 2.749 | 0.945 | – | 0.050 | 1.847 |
| | SB Granular MoE | 2.708 | 0.925 | – | 0.044 | 1.816 |
| | PB Granular MoE | 2.699 | 0.924 | – | 0.046 | 1.811 |
| 400B FineWeb ∪ 200B Stack | Dense Baseline Union | 2.866 | 1.050 | – | – | 1.958 |
| | PB Granular MoE Union | 2.630 | 0.935 | – | – | 1.782 |
| | Dense Baseline | 4.028 | – | 1.279 | 1.399 | 2.654 |
| | SB Switch MoE | 3.810 | – | 1.062 | 1.180 | 2.436 |
| 400B FineWeb → 200B German 0% Replay | PB Switch MoE | 3.782 | – | 1.059 | 1.152 | 2.420 |
| | SB Granular MoE | 3.701 | – | 1.038 | 1.071 | 2.369 |
| | PB Granular MoE | 3.685 | – | 1.028 | 1.055 | 2.356 |
| | Dense Baseline | 2.946 | – | 1.367 | 0.066 | 2.157 |
| | SB Switch MoE | 2.749 | – | 1.142 | 0.039 | 1.946 |
| 400B FineWeb → 200B German 40% Replay | PB Switch MoE | 2.741 | – | 1.129 | 0.042 | 1.935 |
| | SB Granular MoE | 2.701 | – | 1.118 | 0.037 | 1.910 |
| | PB Granular MoE | 2.690 | – | 1.099 | 0.037 | 1.895 |
| 400B FineWeb ∪ 200B German | Dense Baseline Union | 2.938 | – | 1.390 | – | 2.164 |
| | PB Granular MoE Union | 2.669 | – | 1.120 | – | 1.895 |

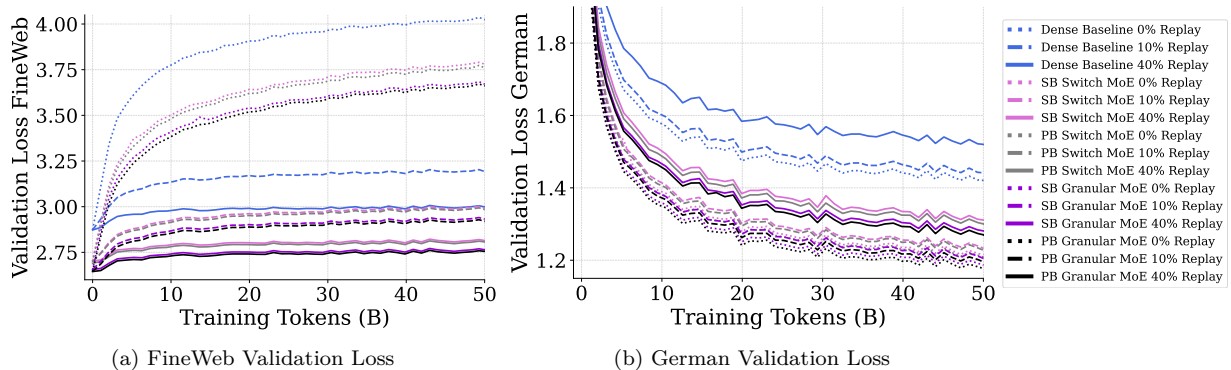

(a) FineWeb Validation Loss

(b) German Validation Loss

Figure 6: **Penalty-Balanced (PB) and Sinkhorn-Balanced (SB) Top-$k$ MoEs behave similarly to the FLOP-matched Dense baseline when being continually pre-trained with varying amounts of replay.** We continually pre-train MoEs and a dense baseline using varying amounts of replay: 0% (dotted curves), 10% (dashed curves), and 40% (full curves). We observe that replay substantially reduces forgetting for all models while slightly harming adaptation; that is, the effect of replay is the same for MoEs as for dense models.

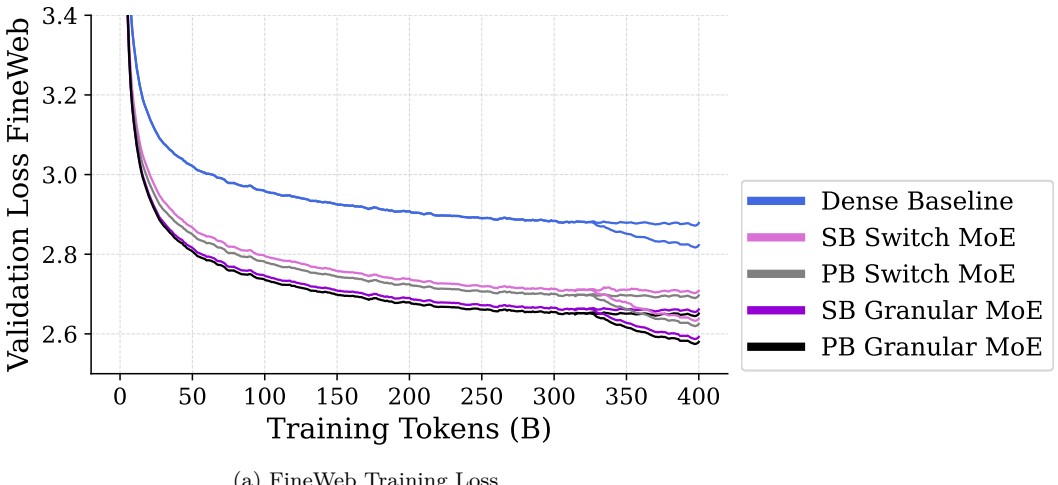

(a) FineWeb Training Loss

Figure 7: **Validation loss during initial pre-training on FineWeb with Infinite LR schedules.** We report decay and constant phases to completion. We observe that all MoE transformers stably decrease validation loss throughout pre-training, with MoEs improving over the dense model as expected. Interestingly, the PBT$k$ MoEs shows an incremental improvement over SBT$k$.

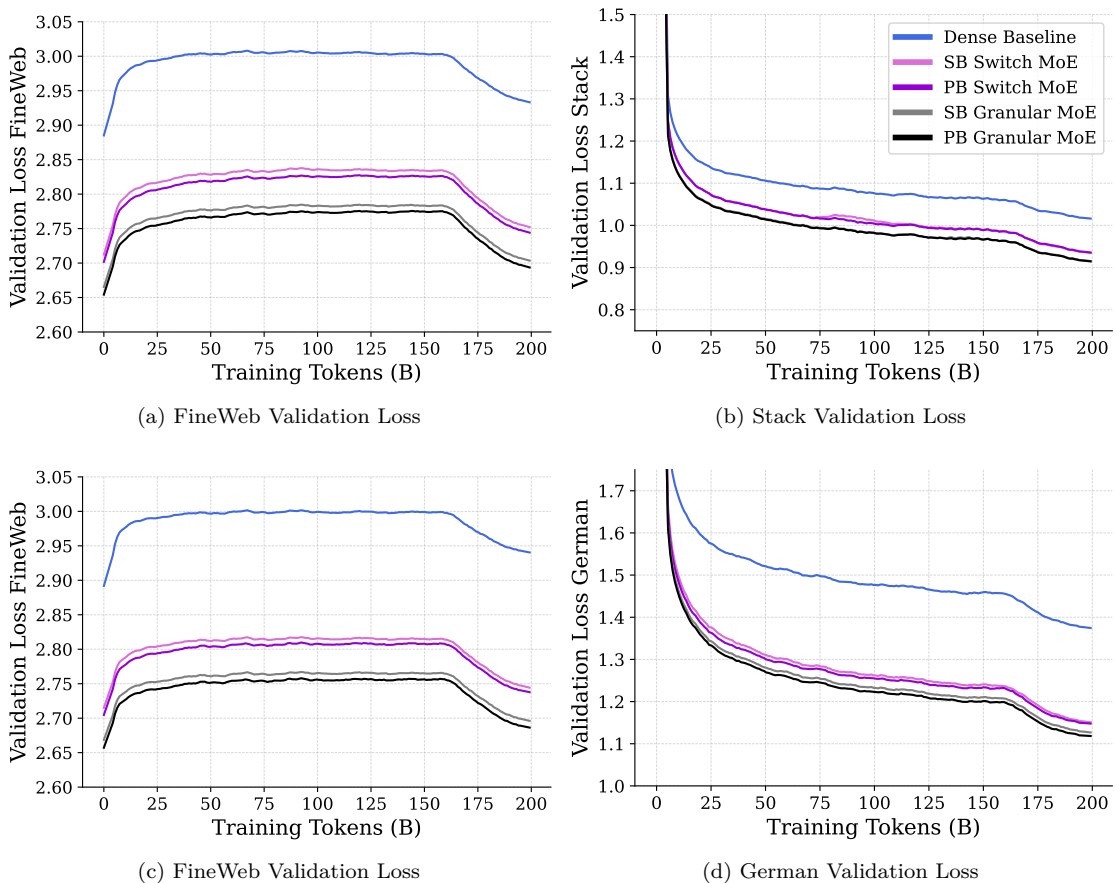

(a) FineWeb Validation Loss

(b) Stack Validation Loss

(c) FineWeb Validation Loss

(d) German Validation Loss

Figure 8: **Validation Loss on CPT and PT datasets during CPT.** Subfigures (a) and (c) report FineWeb validation loss, while subfigures (b) and (d) report Stack and German validation loss respectively for models trained on those datasets. We observe that all MoEs maintain their sample efficiency after the distribution shift, reaching a lower loss in many fewer iterations than the FLOP-matched dense baseline.

### D.3 Qualitative analysis

In the following section, we present new metrics for analyzing the routing decisions of MoEs in our study and interpret how they change during continual pre-training. To accomplish this, we take checkpoints before and after continual pre-training and record their routing decisions, loss, routing imbalance, and a number of different metrics on 20M tokens of FineWeb test data (pre-training dataset), 20M tokens of German test data (continual pre-training dataset), and 20M tokens of Stack test data (continual pre-training dataset). Our results can be grouped into four main categories: 1) routing saturation analysis, 2) vocabulary specialization analysis, 3) expert co-activation analysis, and 4) routing imbalance analysis.

#### D.3.1 Continual routing saturation analysis

We adapt the analysis of router saturation from Muennighoff et al. (2024) to the continual setting. Note that we will directly reproduce and slightly modify some lines from Muennighoff et al. (2024)'s definition of router saturation below for clarity and ease of passing from one paper's notation to the other. Concretely, we define continual `Continual Router Saturation` as:

$$\texttt{Continual Router Saturation}(t, h, j) = \frac{1}{N} \sum_{i=1}^{N} \frac{|\mathcal{E}_i^{(\mathcal{T}_h)} \cap \mathcal{E}_i^{(\mathcal{T}_j)}|}{k}, \tag{2}$$

where:

- $\mathcal{T}_h$ and $\mathcal{T}_j$: The tasks being considered when selecting checkpoints. Note that $h \leq j$. In our case, $j, h \in \{0, 1, 2\}$, with $\mathcal{T}_0$ designating the pre-training task (FineWeb) and $\mathcal{T}_1, \mathcal{T}_2$ designating German and Stack continual pre-training tasks, respectively. While our experiments only consider one transition, in general, there may be many more.

- $N$: The total number of tokens in the dataset.

- $k$: The number of experts activated per input token.

- $\mathcal{E}_i^{(\mathcal{T}_h)}$: The set of $k$ experts activated for the $i$th token at the final checkpoint of the $h$th task.

- $\mathcal{E}_i^{(\mathcal{T}_j)}$: The set of $k$ experts activated for the $i$th token at the final checkpoint of the $j$th task.

- $|\mathcal{E}_i^{(\mathcal{T}_h)} \cap \mathcal{E}_i^{(\mathcal{T}_j)}|$: The number of common experts activated for the $i$th token between the final checkpoints taken from the $h$th task and $j$th task.

Figures 9 and 10 consider $h = 0$ and $j \in \{0, 1\}$ for Granular (31 routed experts, 3 active, 1 shared) and Switch (8 routed experts, 1 active) MoEs respectively, thus comparing the checkpoint before continual pre-training with the checkpoint obtained afterward. The subfigures on the right report router saturation across model layers for PBT$k$ MoEs, while the subfigures on the left report the same for switch SBT$k$ MoEs. Each row reports router saturation on a different dataset. We make the following observations:

(1) the first few layers are consistently among those with the lowest router saturation,

(2) router saturation is lower for checkpoints trained on a given task $h$ when it is measured with respect to tokens from $h$, and

(3) the router saturation of models tested on their continual pre-training dataset seems to consistently decrease with a small slope as the layers index increases.

Observation (1) suggests that the early layers may undergo the most change during continual pre-training. Note that the trend of the first few layers having low router saturation is especially pronounced in subfigures (a) and (b), suggesting that most of the forgetting seen during continual pre-training may occur in the early layers. Observation (2) shows that MoEs change their routing decisions more to the distribution they are being trained on in the case of German and Stack, which is intuitive. Observation (3) suggests that layers closer to the final layer of the MoE must change more to adapt to the new distribution.

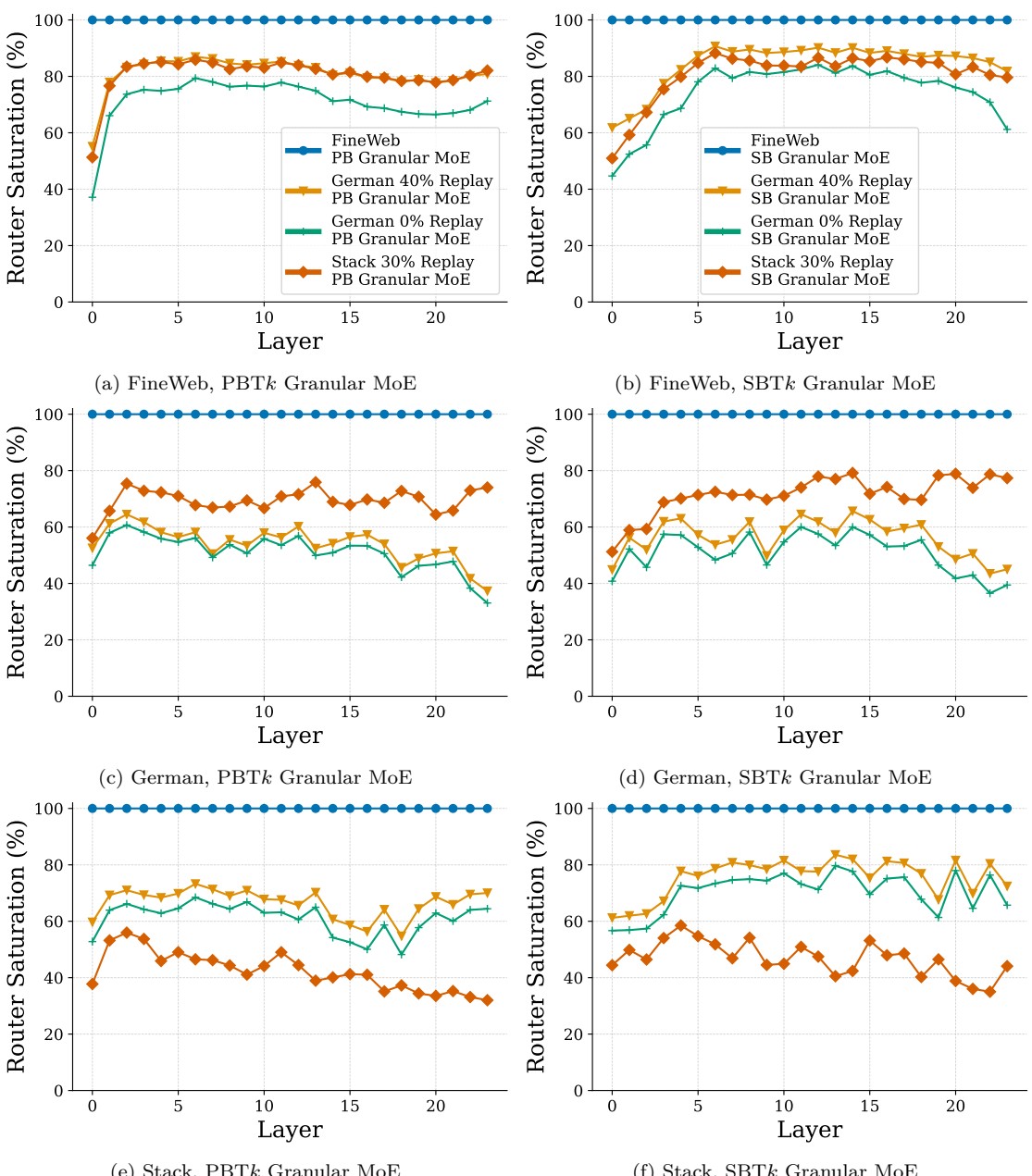

Figure 9: **Router saturation at the beginning of continual pre-training for Granular MoEs.** Subfigures (a,c,e) report layer-wise router saturation for PBT*k* MoEs, while subfigures (b,d,f) report router saturation for SBT*k* MoEs. (a) and (b) measure routing saturation with respect to test tokens from FineWeb, (c) and (d) measure router saturation with respect to test tokens from German, and (e) and (f) measure router saturation with respect to test tokens from Stack. We observe a few trends: 1) the first few layers are consistently among those with the lower router saturation, 2) router saturation is consistently lower for checkpoints CPT on the testing distribution showing that these checkpoitns adapt more to that distribution, 3) the router saturation of models tested on their continual pre-training dataset seems to consistently decrease with a small slope as the layers index increases, and 4) the no-replay checkpoint consistently has lower router saturation than its 40% replay counterpart.

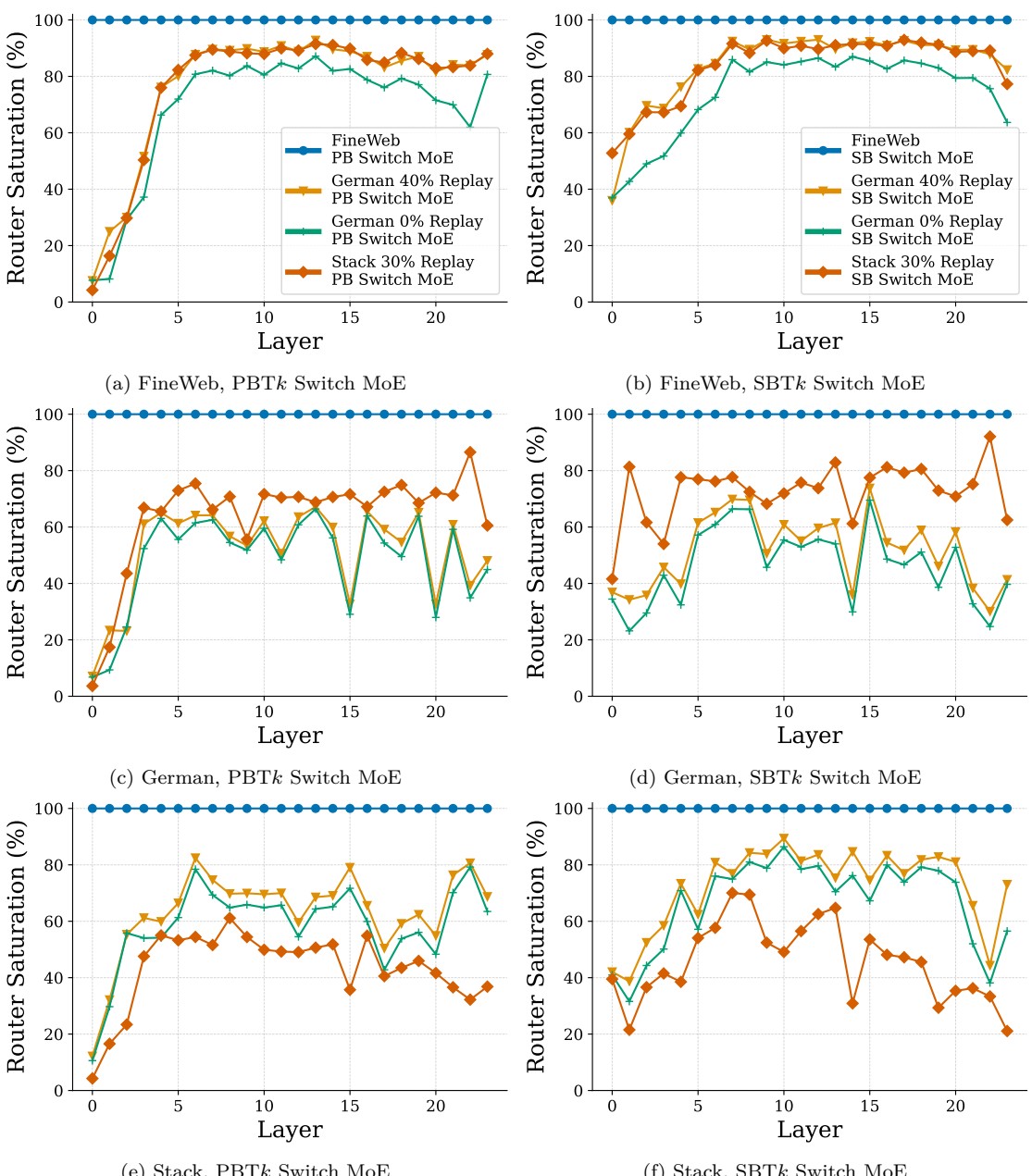

Figure 10: **Router saturation at the beginning of continual pre-training for Switch MoEs.** Subfigures (a,c,e) report layer-wise router saturation for PBT*k* MoEs, while subfigures (b,d,f) report router saturation for SBT*k* MoEs. (a) and (b) measure routing saturation with respect to test tokens from FineWeb, (c) and (d) measure router saturation with respect to test tokens from German, and (e) and (f) measure router saturation with respect to test tokens from Stack. We observe a few trends: 1) the first few layers are consistently among those with the lower router saturation, 2) router saturation is consistently lower for checkpoints CPT on the testing distribution showing that these checkpoints adapt more to that distribution, 3) the router saturation of models tested on their continual pre-training dataset seems to consistently decrease with a small slope as the layers index increases, and 4) the no-replay checkpoint consistently has lower router saturation than its 40% replay counterpart.

### D.3.2 Continual vocabulary specialization analysis

We adapt the analysis of vocabulary specialization from Muennighoff et al. (2024) to the continual setting. Note that we will directly reproduce and slightly modify some lines from Muennighoff et al. (2024)'s definition of vocabulary specialization below for clarity and ease of passing from one paper's notation to the other. Concretely, we define `Vocabulary Specialization` as:

$$\text{Vocabulary Specialization}(j, E_i, x) = \frac{N_{j,x,E_i}^{(k)}}{N_{j,x}}, \tag{3}$$

where:

- $E_i$: The $i$th expert in an MoE layer.

- $j$: A task index specifying which final checkpoint to use (e.g., specifying the final checkpoint after task 1, task 2,...).

- $x$: The token ID being analyzed.

- $k$: The number of experts considered (we use $k=3$ for Granular MoEs and k=1 for switch MoEs).

- $N_{j,x,E_i}^{(k)}$: The number of times input data is routed to $E_i$ for $x$ when using the final checkpoint of task $j$.

- $N_{j,x}$: The total number of times input data is routed across all experts for $x$ and the final checkpoint of task $j$.

`Vocabulary Specialization` can, therefore, be calculated for each expert at every layer of the model and for each token in the model's vocabulary. By assigning each token in the vocabulary to the expert that processes it the most frequently, we can then create a *one-to-many mapping* between experts and vocabulary entries for each layer of the MoE. Then, we can calculate the average vocabulary specialization of each expert by averaging over its assigned tokens and averaging across experts to measure specialization within a layer. To compare specialization across model checkpoints, we can re-use the *one-to-many mapping* of a previous checkpoint and measure how the specialization with respect to this mapping has changed during continual pre-training. Concretely, the continual vocabulary specialization (`CVS`) for an MoE layer $l$ can be defined as follows:

$$\text{CVS}(j, h) = \frac{1}{N_E} \sum_{x \in \mathcal{V}} \text{Vocabulary Specialization}(h, E_{\alpha_{j,x}}, x) \tag{4}$$

$$\alpha_{j,x} := \underset{i \in [N_E]}{\arg\max} \left\{ \text{Vocabulary Specialization}(j, E_i, x) \right\} \tag{5}$$

- $N_E$: The number of experts in an MoE layer $l$.

- $\mathcal{V}$: The set of tokens in the model's vocabulary (we use the Llama3 tokenizer).

- $h$: A task index specifying the checkpoint from which to compute the mapping.

- $j$: A task index specifying a final checkpoint that is used to compute the continual vocabulary specialization.

Note that the dataset of tokens used to compute the `CVS` is omitted for simplicity. However, the specialization of experts will depend on the distribution of the tokens because the same input token may be routed to different experts depending on the context within which it lives and the context will change depending on the distribution. For instance, the hidden representation of the word "for" in an English language corpus and a code corpus may differ wildly.

Figures 11 and 12 report the `CVS` of Granular and Switch MoEs, respectively. For each plot, the *one-to-many mapping*, $\alpha_{j,x}$, is created from the checkpoint pre-trained on FineWeb (e.g., the checkpoint we start continual pre-training from). All specializations are computed with respect to the input token. When evaluated on FineWeb, we observe across all architectures and balancing strategies that the first few layers for continually pre-trained models have lower continual vocabulary specialization than the pre-trained checkpoint, whereas subsequent layers have vocabulary specialization that closely matches that of the pre-trained checkpoint. This is even the case for the model that uses 0% replay, suggesting that MoEs learn routing policies during pre-training that are relatively unaffected by continual pre-training. When evaluated on German and stack, we observe that all MoEs continually pre-trained on those datasets have lower vocabulary specialization than models not trained on those distributions, showing their adaptation. On German, the zero-replay model has the smallest CVS. We hypothesize that this is the case because these models adapt the most to the German distribution and happen to learn new routing patterns, distinct from the ones used on FineWeb. Contrasting the results observed in subfigures (a) and (b) across Figures 11 and 12 to other subfigures, we observe that the vocabulary specialization on the pre-training dataset only changes for the first few layers, while it changes across all layers for the data seen during continual pre-training, even for the model that does not utilize any replay. Contrasting this with the stronger performance of the no-replay model on German and its poorer performance on FineWeb, the superior adaptation to German is correlated to the change in vocabulary specialization throughout the model while the poorer performance on the previous distribution is correlated with larger changes in vocabulary specialization in the first few layers.

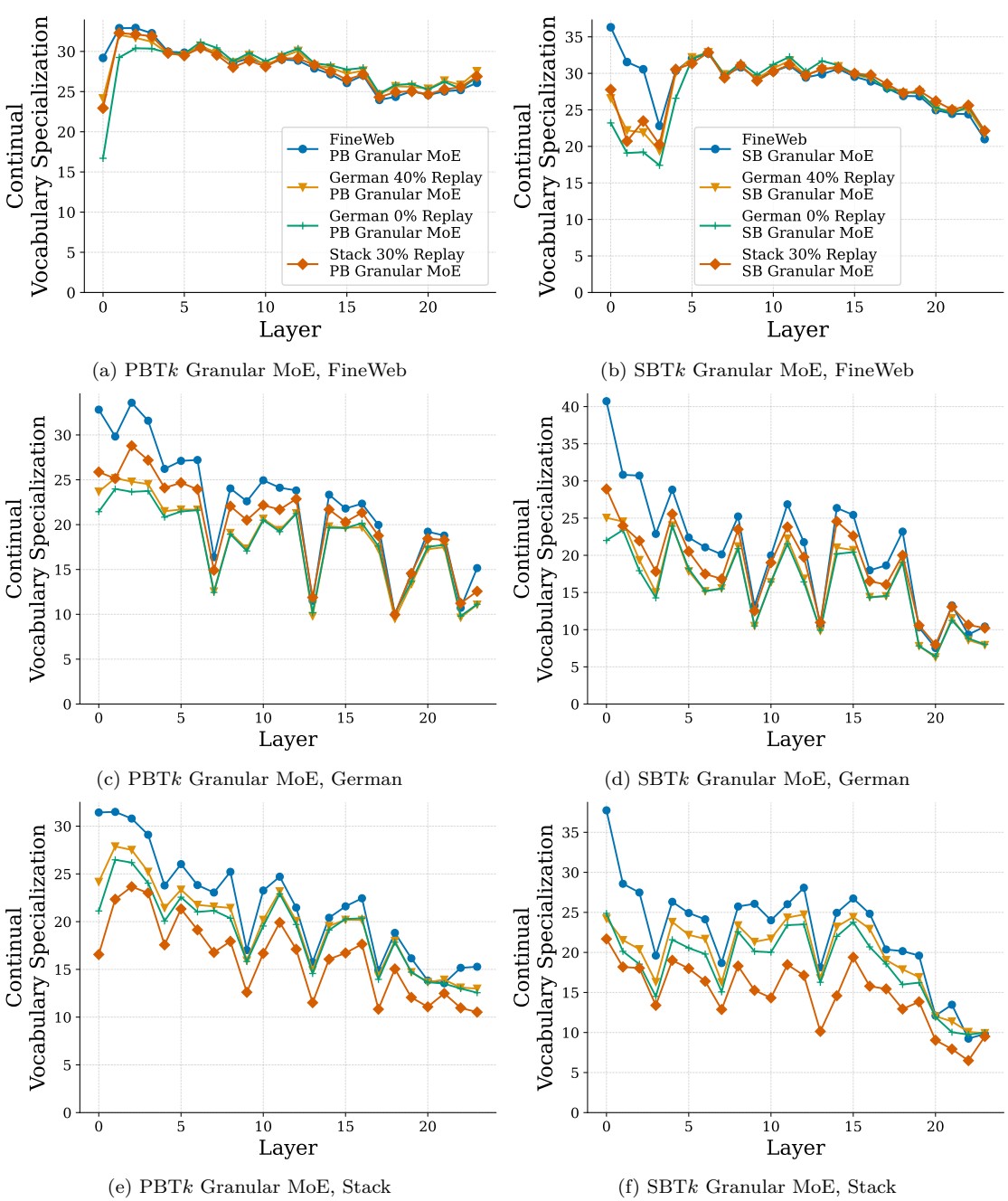

Figure 11: **Continual Vocabulary Specialization for Granular MoEs.** We report `CVS` for each MoE layer in the MoEs when testing models on FineWeb, German, and Stack. We observe that early layers deviate most from the checkpoint after pre-training, while later layers in the continually pre-trained MoEs nearly match the vocabulary specialization of their checkpoints after the first phase pre-training. This is even the case for the checkpoint that does not replay previous data, suggesting that vocabulary specialization for pre-training data is mostly determined during the initial pre-training phase.

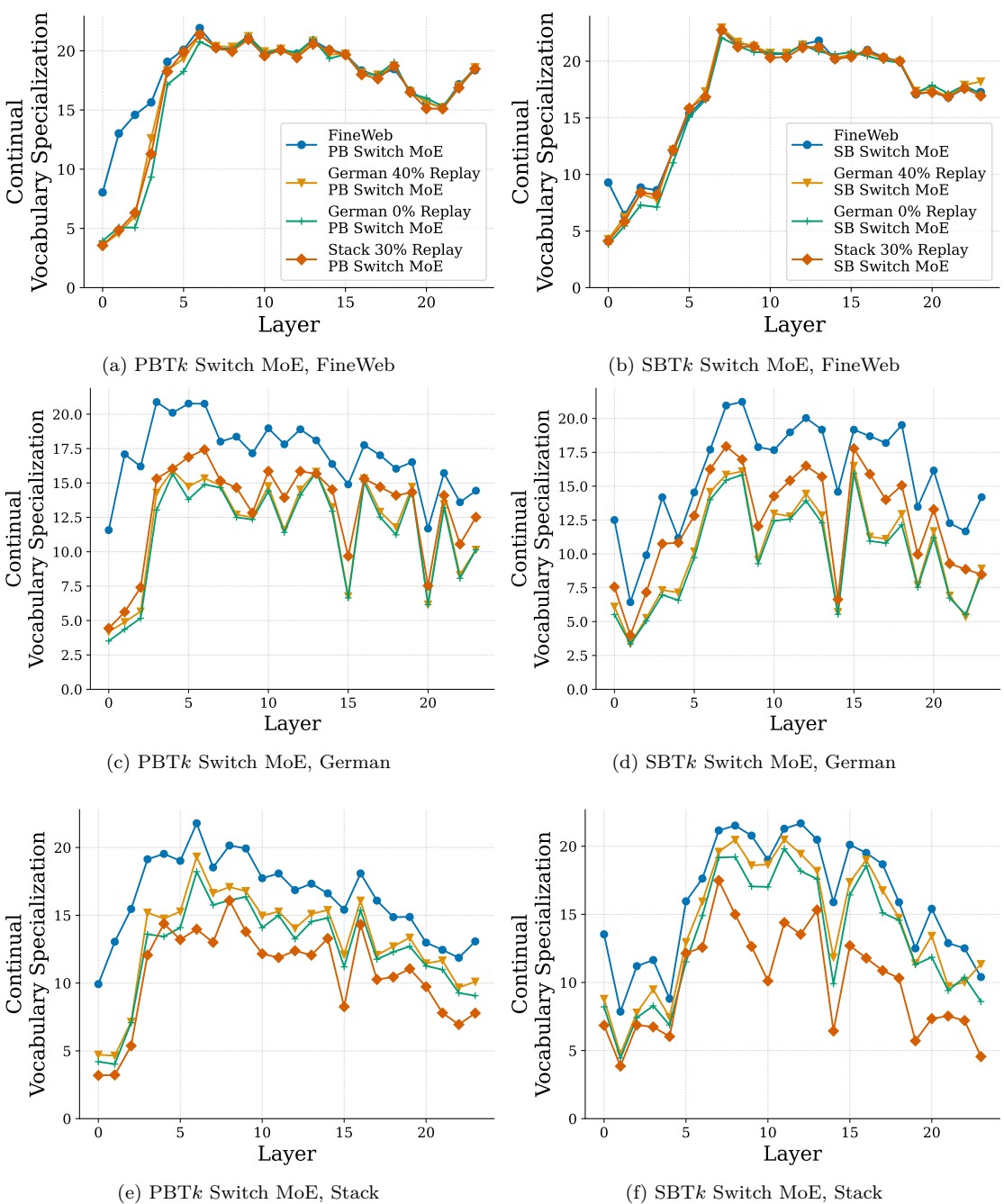

(a) PBT*k* Switch MoE, FineWeb

(b) SBT*k* Switch MoE, FineWeb

(c) PBT*k* Switch MoE, German

(d) SBT*k* Switch MoE, German

(e) PBT*k* Switch MoE, Stack

(f) SBT*k* Switch MoE, Stack

Figure 12: **Continual Vocabulary Specialization for Switch MoEs.** We report `CVS` for each MoE layer in the MoEs when testing models on FineWeb, German, and Stack. We observe that early layers deviate most from the checkpoint after pre-training, while later layers in the continually pre-trained MoEs nearly match the vocabulary specialization of their checkpoints after the first phase pre-training. This is even the case for the checkpoint that does not replay previous data, suggesting that vocabulary specialization for pre-training data is mostly determined during the initial pre-training phase.

### D.3.3 Continual expert co-activation analysis

We adapt the analysis of expert co-activation from Muennighoff et al. (2024) to the continual setting. Note that we will directly reproduce and slightly modify some lines from Muennighoff et al. (2024)'s definition of expert co-activation below for clarity and ease of passing from one paper's notation to the other. Concretely, we define `Expert Co-activation` as:

$$\texttt{Expert co-activation}(E_i, E_j) = \frac{N_{E_i, E_j}}{N_{E_i}}, \tag{6}$$

where:

- $E_i$: The first expert.

- $E_j$: The second expert.

- $N_{E_i, E_j}$: The number of times experts $E_i$ and $E_j$ are activated together.

- $N_{E_i}$: The total number of times expert $E_i$ is activated.

The co-activation matrix $C$ for any layer in the MoE can, therefore, be created by setting $C_{i,j} = $ `Expert co-activation`$(E_i, E_j)$. Then, we can define the co-activation difference as follows:

$$\texttt{Co-activation Difference}(p, q) = |C^{(p)} - C^{(q)}|. \tag{7}$$

Where $|\cdot|$ is the coordinate-wise absolute value function. Each coordinate $i, j$ of the co-activation difference measures the change in expert co-activation for experts $i, j$ between final MoE checkpoints after tasks $p$ and $q$, respectively. Taking statistics of the entries of the co-activation difference matrix allows us to measure how expert co-activation changes globally at each layer during continual pre-training.

In Figure 13, we report the median of the coordinates of the co-activation difference matrix between each continually pre-trained Granular MoE in our study (the switch MoEs only activate a single expert so they have no co-activation) and its checkpoint after the initial pre-training phase. We observe that when evaluated on the FineWeb test set, the Penalty-Balanced MoEs have the largest median differences overall and that they are most pronounced in the first two layers and layer 18. When evaluated on the German test set, we observe that the models continually pre-trained on German have the largest median differences and that the tendency for Penalty-Balanced MoEs to have large differences is maintained. When evaluated on the Stack test set, similar trends are observed.

In Figures 14 and 15, we visualize a subset of the full expert co-activation matrices for Penalty-Balanced and Sinkhorn-Balanced MoEs, respectively. Specifically, we show expert co-activations for the 16 experts with the largest co-activation values. The left-most plots show the co-activation matrix of the checkpoint continually pre-trained on FineWeb without decaying, the middle plots show the co-activation matrix of the checkpoint after continually pre-training on Stack, and the rightmost figures show the co-activation difference matrix. Subfigure (a) shows layer 0, (b) shows layer 11, and (c) shows the final layer. We observe that the co-activation difference is the largest for layer 0 for both Penalty-Balanced and Sinkhorn-Balanced MoEs. Notably, for the Sinkhorn-Balanced granular MoE's pre-trained checkpoint, most of the co-activation weight is placed on the expert 15. However, this strong weighting on expert 15 is attenuated during continual pre-training. In contrast, the co-activations are more dispersed in the Penalty-Balanced MoE. For layers 11 and 23, there is minimal change between pre-training and continual pre-training.

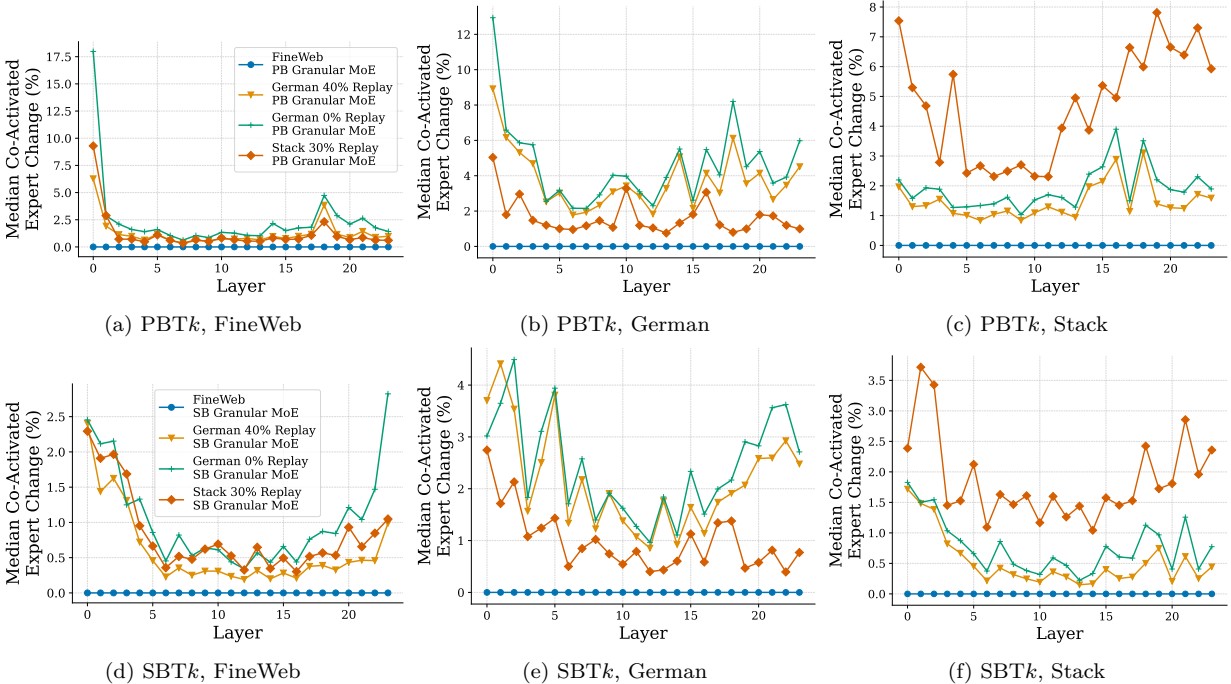

Figure 13: **Layer-wise Median Router Co-activation Difference for Granular MoEs.** We report the median of the coordinates of the co-activation difference matrix between each model in the legend and its corresponding pre-trained checkpoint. We observe that on FineWeb, the Penalty-Balanced MoEs have the largest median differences overall and that they are most pronounced in the first two layers and layer 18. On German, we observe that the models continually pre-trained on German have the largest median differences and that the tendency for Penalty-Balanced MoEs to have large differences is maintained. On Stack, similar trends are observed.

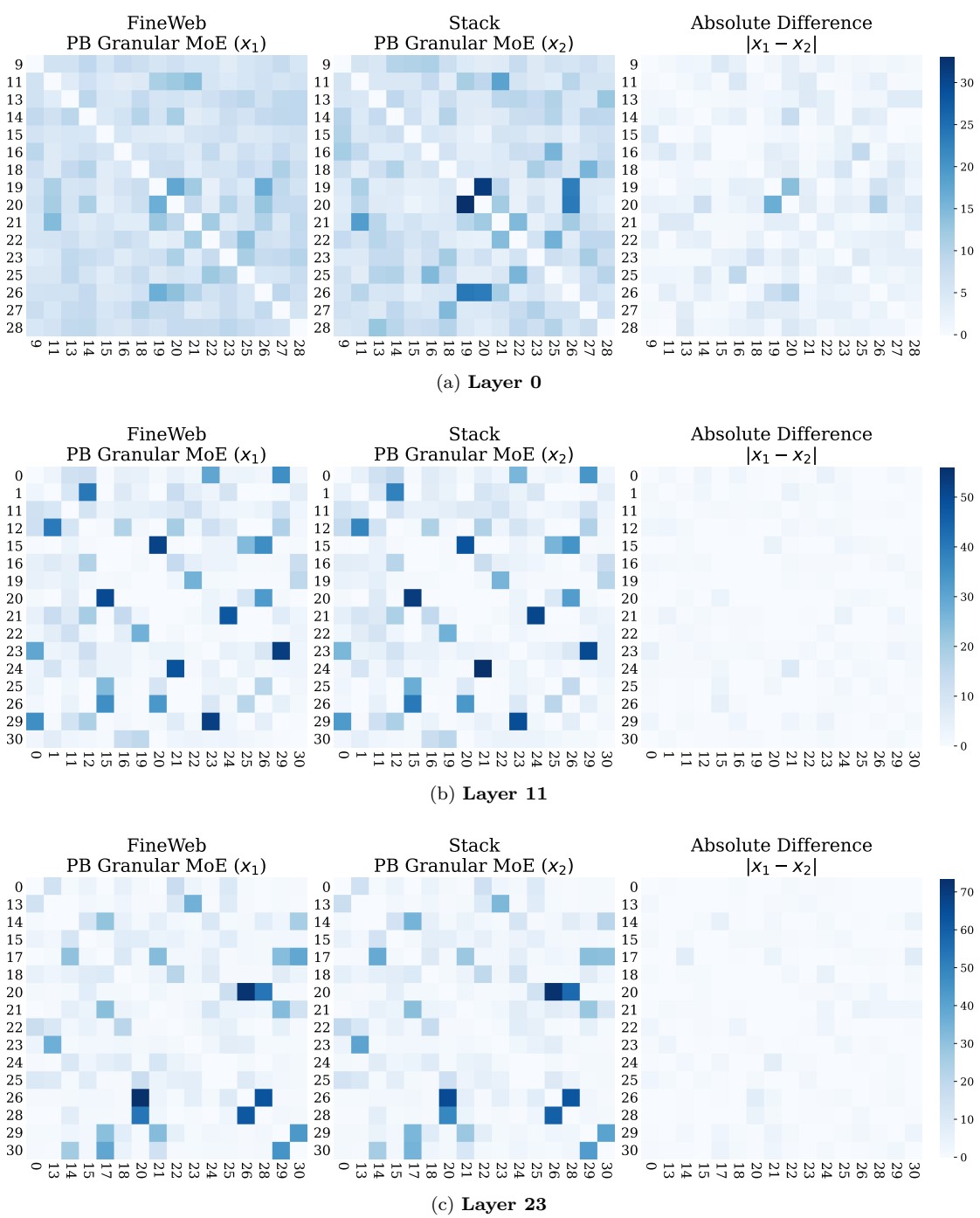

Figure 14: **FineWeb Router Co-activation Matrix for PB Granular MoE Continually Pre-trained on Stack.** The left-most plots show the Co-activation matrix of the checkpoint continually pre-trained on FineWeb without decaying, the middle plots show the Co-activation matrix of the checkpoint after continually pre-training on Stack, and the rightmost figures show the co-activation difference matrix. Subfigure (a) shows layer 0, (b) shows layer 11, and (c) shows the final layer. We observe that the co-activation difference is the largest layer 0, while the other layers change minimally.

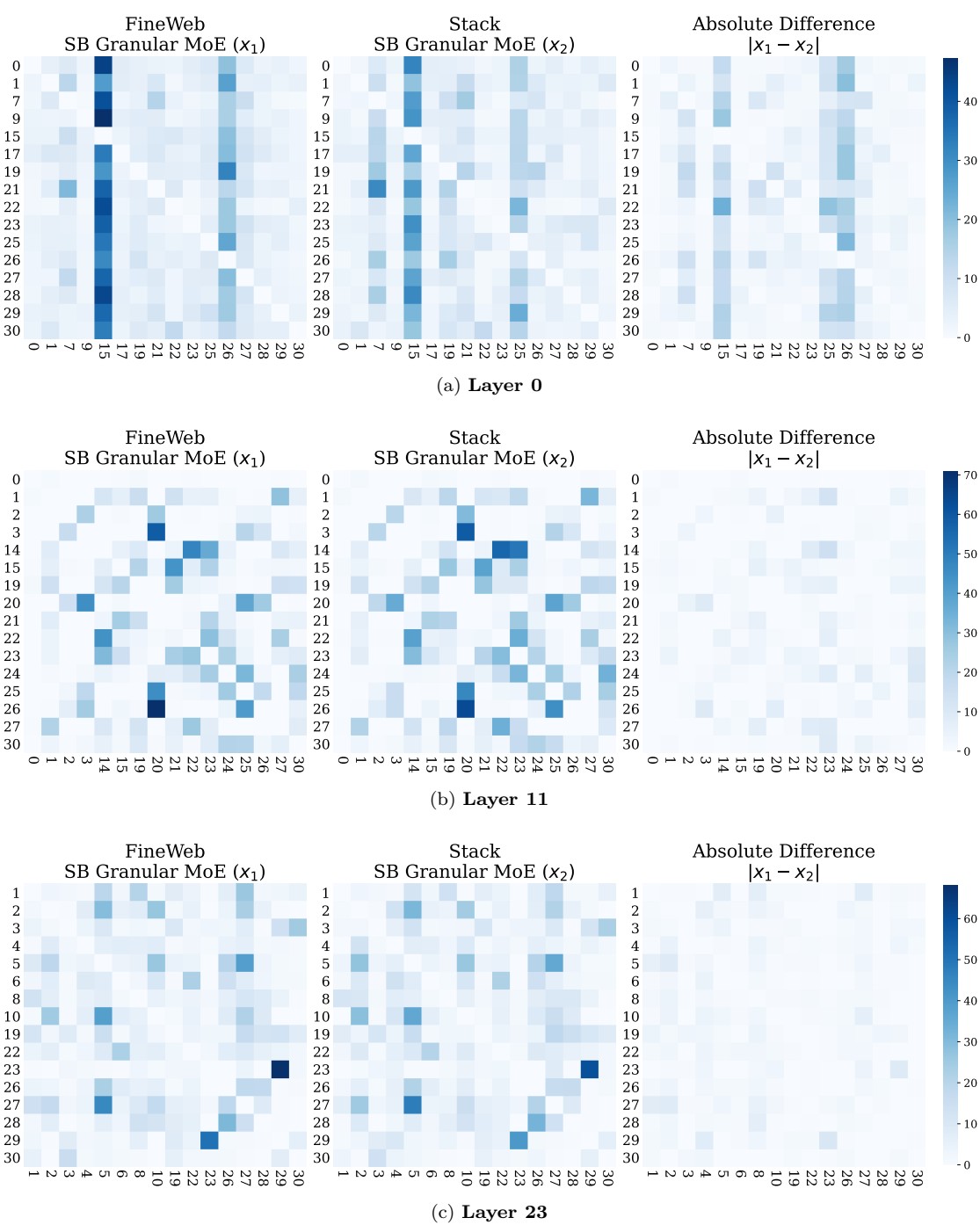

Figure 15: **FineWeb Router Co-activation Matrix for SB Granular MoE Continually Pre-trained on Stack.** The left-most plots show the Co-activation matrix of the checkpoint continually pre-trained on FineWeb without decaying, the middle plots show the Co-activation matrix of the checkpoint after continually pre-training on Stack, and the rightmost figures show the co-activation difference matrix. Subfigure (a) shows layer 0, (b) shows layer 11, and (c) shows the final layer. We observe that the co-activation difference is the largest in layer 0, with most of the weight placed on expert 15. We observe that this strong weighting on expert 15 is attenuated during continual pre-training. For other layers, there is minimal change in co-activation between pre-training and continual pre-training.

### D.3.4 Continual routing imbalance analysis

While performance is one important axis of robustness to distribution shifts, maintaining a balanced load across experts is just as important for MoE foundation models. Without a balanced load, MoE transformers inferenced using expert parallelism without token dropping (e.g., as is done for SOTA models (DeepSeek-AI et al., 2025b; Zhao et al., 2025)) could be bottlenecked by the speed of a single accelerator that receives all the tokens, leading to underutilization of the hardware, lower throughput, and higher costs. To quantitatively assess the effect of distribution shift on load balance, we propose the maximum routing imbalance (MRI): the largest proportion of tokens routed to a single expert in a given MoE layer. Concretely, the MRI at a training iteration $t$ and MoE layer $j$ is defined as

$$\mathrm{MRI}(t, j) := \max_{i \in [1,\dots,E]} \left[ \frac{\sum_{x \in B} \mathbb{1}\{i \in I_k(x)\}}{|B|} \right]. \tag{8}$$

Where $B$ is a set containing all tokens in a given batch, $\mathbb{1}$ is the indicator function, $E$ is the number of routed experts, and $k$ is the number of active experts. *Since latency increases with computation, and, in an MoE layer, the computation required by a given device increases with the load of experts on that device, then MRI calculated with respect to routing decisions on a distribution is a proxy for the worst case latency of an MoE layer on the distribution.* We will use the MRI throughout the following sections to measure the effect of algorithmic changes to continual pre-training on routing imbalance.

In Figures 16 and 17, we set $t$ to be the final iteration of training for each model during pre-training and continual pre-training where it is applicable. The figures plot the layer identity on the x-axis and the MRI on the y-axis. The left column plots report the MRI of PBT$k$ MoEs, while the right column plots report MRI for SBT$k$ MoEs. Figures 16 shows Granular MoEs, while Figure 17 shows switch MoEs. For Granular MoEs, we observe that the MRI for Penalty-Balanced MoEs is consistently lower than for Sinkhorn-Balanced MoEs, that little increase in MRI on FineWeb is incurred during continual pre-training, even for the 0% replay model, and that MoEs become most unbalanced when seeing out-of-distribution data (e.g., see non-german models in (b) and non-code models in (c)). For Switch MoEs, we observe the MRI for Penalty-Balanced MoEs is similarly consistently lower than for Sinkhorn-Balanced MoEs, that similar to Granular MoEs little increase in MRI on FineWeb is incurred during continual pre-training, even for the 0% replay model, that switch MoEs become most unbalanced when seeing out-of-distribution data (e.g., see non-german models in (c,d) and non-code models in (e,f)), and that high MRI is prevalent in early layers independent of the training and testing distributions used, unlike for Granular MoEs. Contrasting these differences with the superior language modeling performance of granular MoEs, one could hypothesize that the unstable MRI observed in early layers for switch models that is not present in Granular MoEs may be a cause of the performance difference.

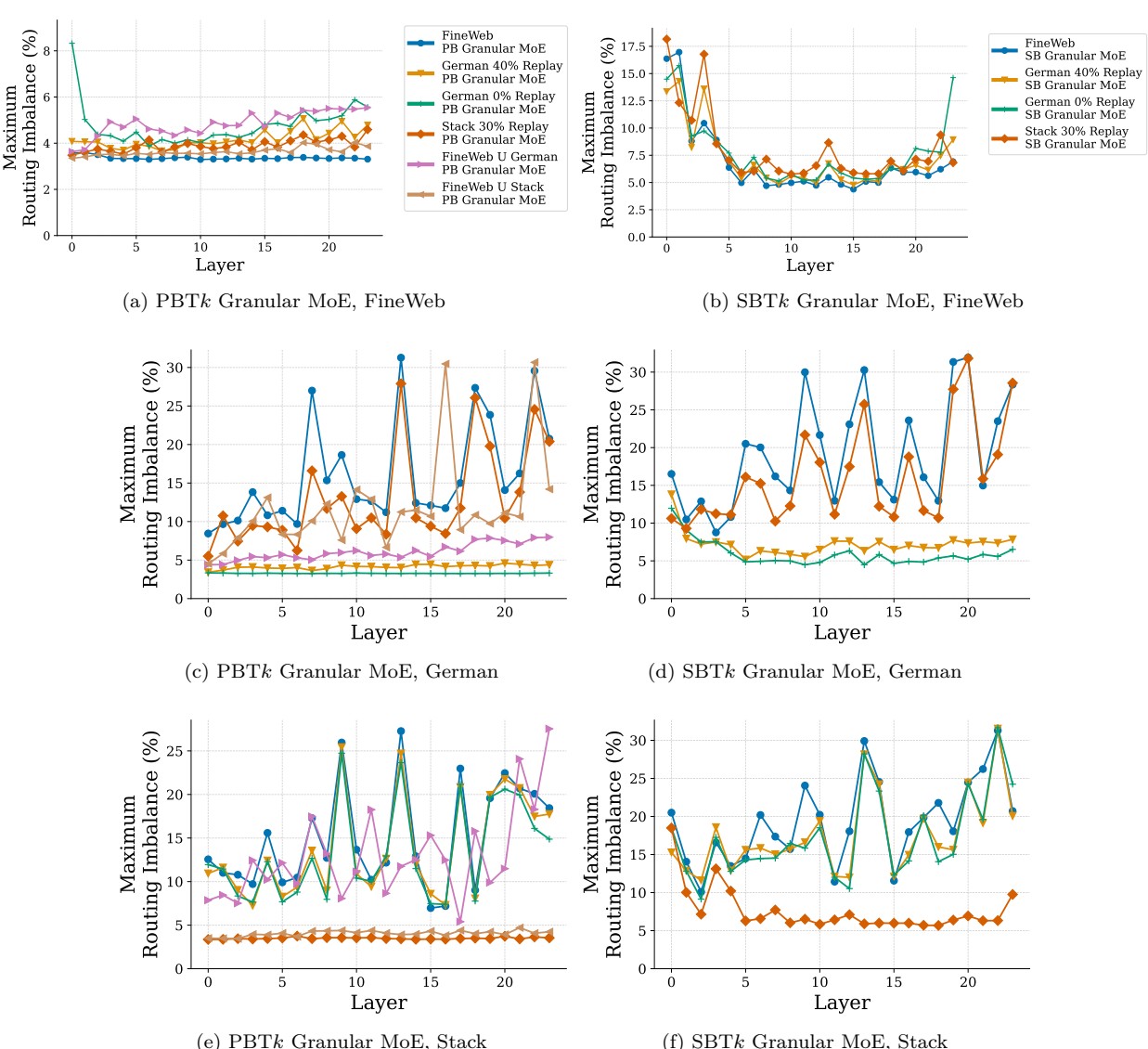

(a) PBT*k* Granular MoE, FineWeb          (b) SBT*k* Granular MoE, FineWeb

(c) PBT*k* Granular MoE, German          (d) SBT*k* Granular MoE, German

(e) PBT*k* Granular MoE, Stack          (f) SBT*k* Granular MoE, Stack

Figure 16: **Layer-wise Maximum Routing Imbalance (MRI) for Granular MoEs.** We report the MRI for each layer in the MoE as a percentage of all routing decisions made on a given dataset's 20M token test set ((a,b)FineWeb, (c,d)German, and (e,f) Stack). The left column PBT*k* MoEs, while the left column reports results for SBT*k* MoEs. We observe that the MRI for Penalty-Balanced MoEs is consistently lower than for comparable Sinkhorn-Balanced MoEs, that little increase in MRI on FineWeb is incurred during continual pre-training, even for the 0% replay model (except for its first layer), and that MoEs become most unbalanced when seeing out-of-distribution data (e.g., see non-german models in (e,f) and non-code models in (c,d)).

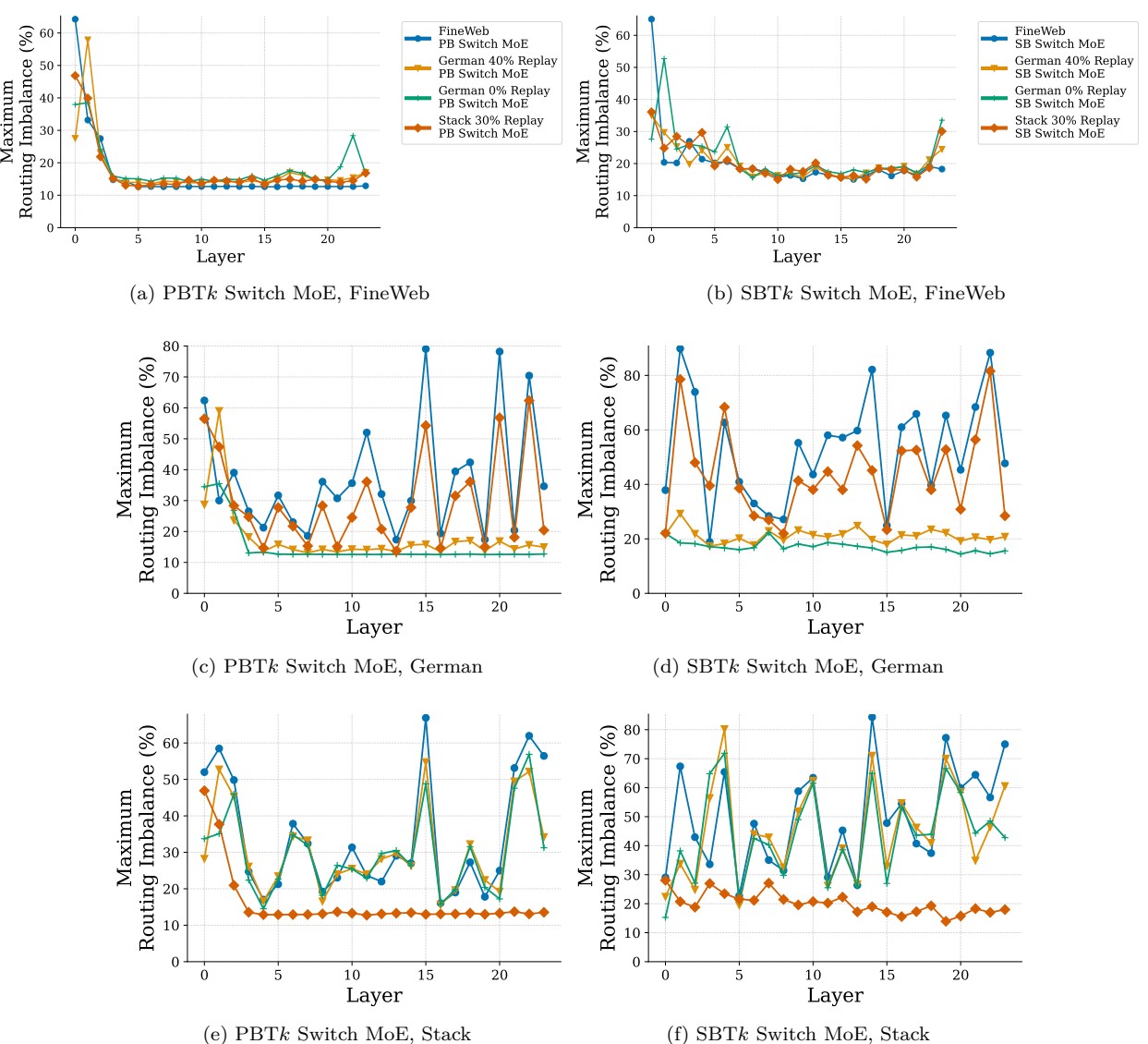

(a) PBT*k* Switch MoE, FineWeb

(b) SBT*k* Switch MoE, FineWeb

(c) PBT*k* Switch MoE, German

(d) SBT*k* Switch MoE, German

(e) PBT*k* Switch MoE, Stack

(f) SBT*k* Switch MoE, Stack

Figure 17: **Layer-wise Maximum Routing Imbalance (MRI) for Switch MoEs.** We report the MRI for each layer in the MoE as a percentage of all routing decisions made on a given dataset's 20M token test set ((a,b) FineWeb, (c,d) German, and (e,f) Stack). The left column PBT*k* MoEs, while the left column reports results for SBT*k* MoEs. We observe that the MRI for Penalty-Balanced MoEs is consistently lower than for comparable Sinkhorn-Balanced MoEs, that little increase in MRI on FineWeb is incurred during continual pre-training, even for the 0% replay model (except for its first layer), that MoEs become most unbalanced when seeing out-of-distribution data (e.g., see non-german models in (e,f) and non-code models in (c,d)), and that high MRI is prevalent in early layers of Switch MoEs independent of the training and testing distributions used.

### D.4 Maximum routing imbalance of MoEs during continual pre-training.

In the following section, we report on the maximum routing imbalance of MoEs during CPT. Specifically, Figures 18 and 19 report routing immediately before and after the distribution shift, while Figures 20 and 21 report MRI during training for Granular and switch MoEs.

**Routing imbalance during training**  By sparsely activating their weight matrices, MoEs experience performance benefits over FLOP-matched dense models. However, this comes at the cost of increased latency when the model's forward pass is bottlenecked by the latency of a single expert. This can become a problem if a router at any layer of the MoE chooses to dispatch a majority of the token load to a particular expert. Therefore, we can estimate the impact of continual pre-training on MoE latency by tracking the worst load imbalance at each layer of the MoE, which is the definition of the maximum routing imbalance equation 1.

In figures 20, and 21, we plot the MRI throughout pre-training (FineWeb) and continual pre-training (German CC) for Switch and Granular MoEs, respectively. Subfigure (a) show the training time and inference time MRI for PB MoEs, while subfigure (b) shows the training time MRI for SB MoEs and subfigure (c) show the inference time MRI for SB MoEs. We distinguish between Sinkhorn Balanced training and inference because the Sinkhorn Balancing algorithm is incompatible with autoregressive generation, so in subfigure (c) we show MRI for SB models without the balancing step (e.g., what would be used during autoregressive generation).

For all MoEs, early layers (0-6) seem to have the largest MRI. For Switch and Granular MoEs alike, we observe that SB routing follows a very similar pattern all throughout pre-training. During the continual pre-training phase, we observe that this pattern changes slightly, actually becoming more balanced throughout continual pre-training for both inference time and training time routing imbalance. Turning our attention to the PB MoEs, we observe that Switch MoEs suffer from much greater routing imbalance than their dense Granular counterparts in early layers. However, for most layers of the PB switch MoE and all layers of the PB Granular MoE, the MRI quickly reaches a smaller value than their SB counterparts during pre-training and continual pre-training showing that PB MoEs are also robust to distribution shifts, but that the Granular MoE architecture is favorable for continual pre-training

*In summary, both PB and SB Top-k routing algorithms are robust to distribution shifts, with PB initially being more perturbed by the distribution shifts, but recovering quickly to a better balance then SB. These results demonstrate that using infinite LR schedules and replay is enough to continually pre-train MoE LLMs without incurring a large increase in MRI.*

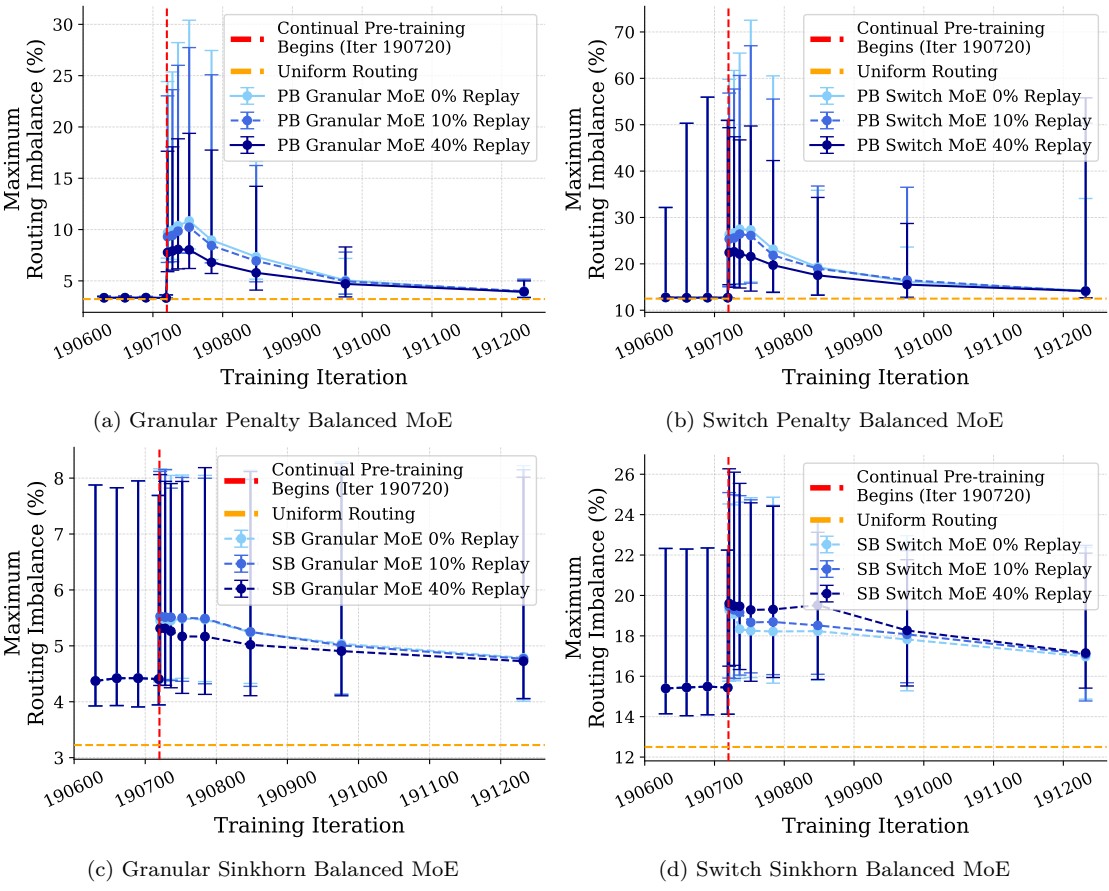

(a) Granular Penalty Balanced MoE

(b) Switch Penalty Balanced MoE

(c) Granular Sinkhorn Balanced MoE

(d) Switch Sinkhorn Balanced MoE

Figure 18: **Sinkhorn-Balanced (SB) and Penalty-balanced (PB) Top-$k$ MoEs show little change in training-time maximum routing imbalance as a result of adjusting the replay percentage.** We report the median MRI observed across MoE layers with min and max error bars shortly before and following the distribution shift when continually pre-training the MoEs on German CC. We observe that independent of the replay percentage used, the MoEs recover pre-training level median MRI within 1000 iterations of continual pre-training. However, replay does mitigate the increase in MRI caused by the distribution shift to a small extent in PB MoEs. In contrast, the SB MoEs are quite robust to the distribution shift and their routing patterns seem to be invariant to the replay percentage used.

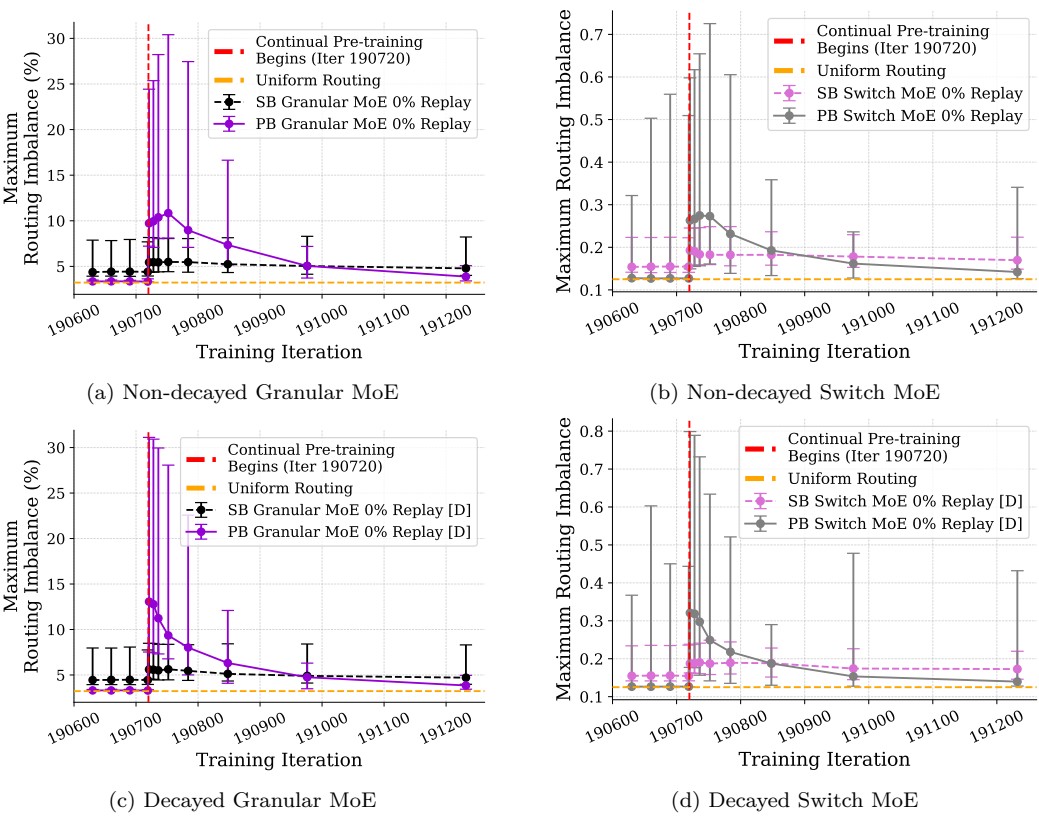

(a) Non-decayed Granular MoE

(b) Non-decayed Switch MoE

(c) Decayed Granular MoE

(d) Decayed Switch MoE

Figure 19: **Decayed Penalty-Balanced (PB) Top-$k$ MoEs have slightly higher MRI during distributions shifts than their non-decayed counterparts.** We report the median MRI observed across MoE layers with min and max error bars shortly before and following the distribution shift when continually pre-training the MoEs on German CC. We observe that all SB MoE keep a stable MRI throughout the distribution shift, showing that they are mostly unaffected. In contrast, the PB checkpoints suffer from strong routing imbalance after the distribution shift but recover quickly, with the decayed checkpoints reaching a marginally higher MRI.

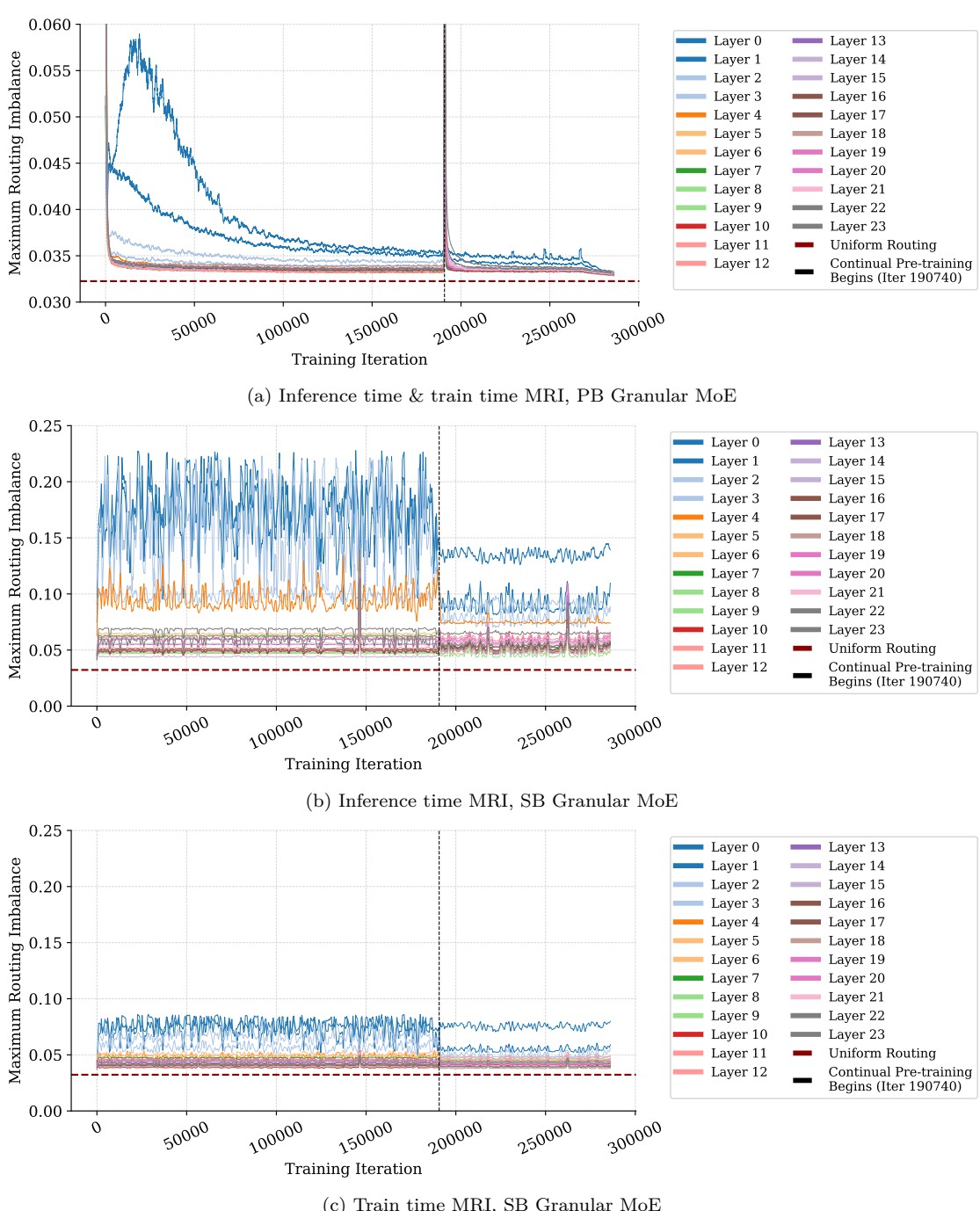

(a) Inference time & train time MRI, PB Granular MoE

(b) Inference time MRI, SB Granular MoE

(c) Train time MRI, SB Granular MoE

Figure 20: **Training time and inference time MRI for Granular MoEs throughout pre-training and continual pre-training.** We show layer-wise maximum routing imbalance during pre-training and continual pre-training. While Penalty-Balanced MoEs have the same routing dynamics at training and inference time, Sinkhorn balancing is incompatible with autoregressive generation, so we show both inference-time and train-time MRI for SB models. We observe that early layers in the MoE consistently have the largest MRI for both PB and SB MoEs, the MRI of PB MoEs is much better behaved, and after the distribution shift, the MRI of SB models becomes more stable.

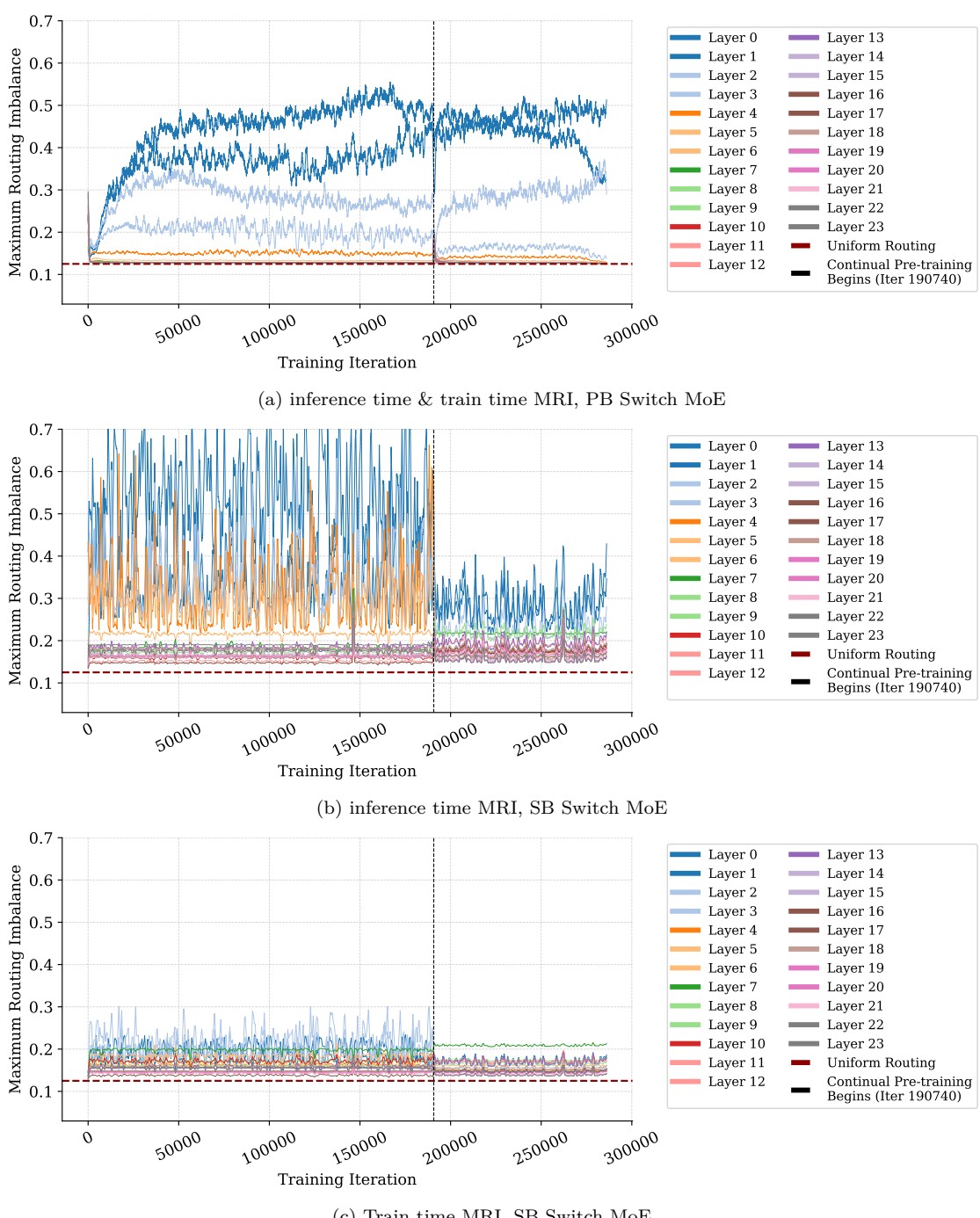

(a) inference time & train time MRI, PB Switch MoE

(b) inference time MRI, SB Switch MoE

(c) Train time MRI, SB Switch MoE

Figure 21: **Training time and inference time MRI for Switch MoEs throughout pre-training and continual pre-training.** We show layer-wise maximum routing imbalance during pre-training and continual pre-training. While Penalty-Balanced MoEs have the same routing dynamics at training and inference time, Sinkhorn balancing is incompatible with autoregressive generation, so we show both inference-time and train-time MRI for SB models. We observe that early layers in the MoE consistently have the largest MRI for both PB and SB MoEs, the MRI of PB MoEs is much better behaved, and after the distribution shift, the MRI of SB models becomes more stable.

# E   Dataset sizes and sampling proportions

In the following section, we report the training tokens used, training dataset sizes, and sampling proportions. Specifically, Table 7 reports the amount of data and its exact composition used for different pre-training phases. Tables 8, 9, and 10 report the amount of training tokens and sampling proportions used for FineWeb, Germand, and Stack, respectively.

Table 7: **Pre-training and Continual Pre-training Tokens.** We report the training tokens for all different model training configurations in this paper. During continual pre-training, each batch contains a proportion of replay tokens from the pre-training dataset and new tokens from the continual pre-training dataset.

| Phase | Training Tokens | New Tokens | Replay Tokens |
|---|---|---|---|
| Pre-training | 400B FineWeb | 400B | – |
| Continual Pre-training | 400B FineWeb → 200B Stack 30% Replay | 140B | 60B |
| | 400B FineWeb → 200B German 40% Replay | 120B | 80B |
| | 400B FineWeb → 200B German 0% Replay | 200B | – |

Table 8: **FineWeb CC: Train, Val, and Test dataset sizes used in our experiments.** For the purposes of our study, we create a more manageable subset of FineWeb by subsampling each Common Crawl dump within FineWeb into smaller subsets. We then sample proportional to the sizes of each subset. We report the full size of the subset we sample from during training (note, we only train on 400B tokens of this subset). The exact sizes and sampling proportions of each split are committed as they span more than a page, but can be made available upon request.

| Source | Train Tokens (B) | Test Tokens (B) | Val. Tokens (B) | Sampling Weight |
|---|---|---|---|---|
| FineWeb CC | 2916.650 | 26.442 | 26.426 | 1.000 |

Table 9: **German CC: Train, Val, and Test dataset sizes used in our experiments.**

| Source | Train Tokens (B) | Test Tokens (B) | Val. Tokens (B) | Sampling Weight |
|---|---|---|---|---|
| German CC | 169.291 | 0.489 | 0.491 | 1.000 |

Table 10: **Stack: Train, Val, and Test dataset sizes used in our experiments.**

| Source | Training Tokens (B) | Test Tokens (B) | Val. Tokens (B) | Sampling Weights |
|---|---|---|---|---|
| YAML | 9.039 | 0.613 | 0.609 | 0.017 |
| Java | 19.730 | 0.587 | 0.587 | 0.174 |
| C | 17.988 | 0.594 | 0.597 | 0.159 |
| Markdown | 21.699 | 0.477 | 0.474 | 0.017 |
| PHP | 16.660 | 0.450 | 0.447 | 0.146 |
| C# | 9.245 | 0.552 | 0.553 | 0.084 |
| JSON | 120.669 | 0.709 | 0.695 | 0.017 |
| TypeScript | 6.892 | 0.418 | 0.414 | 0.063 |
| C++ | 13.998 | 0.538 | 0.539 | 0.124 |
| Python | 15.898 | 0.458 | 0.457 | 0.200 |
| **Total** | 251.819 | 5.396 | 5.372 | 1.000 |

## F   Model hyperparameters

The following section outlines the hyperparameters used to train the MoEs and dense transformers in our study. Specifically, Table 11 reports the hyperparameters of the schedules and Table 12 reports the model hyperparameters. We also show an example infinite learning rate schedule in Figure 22.

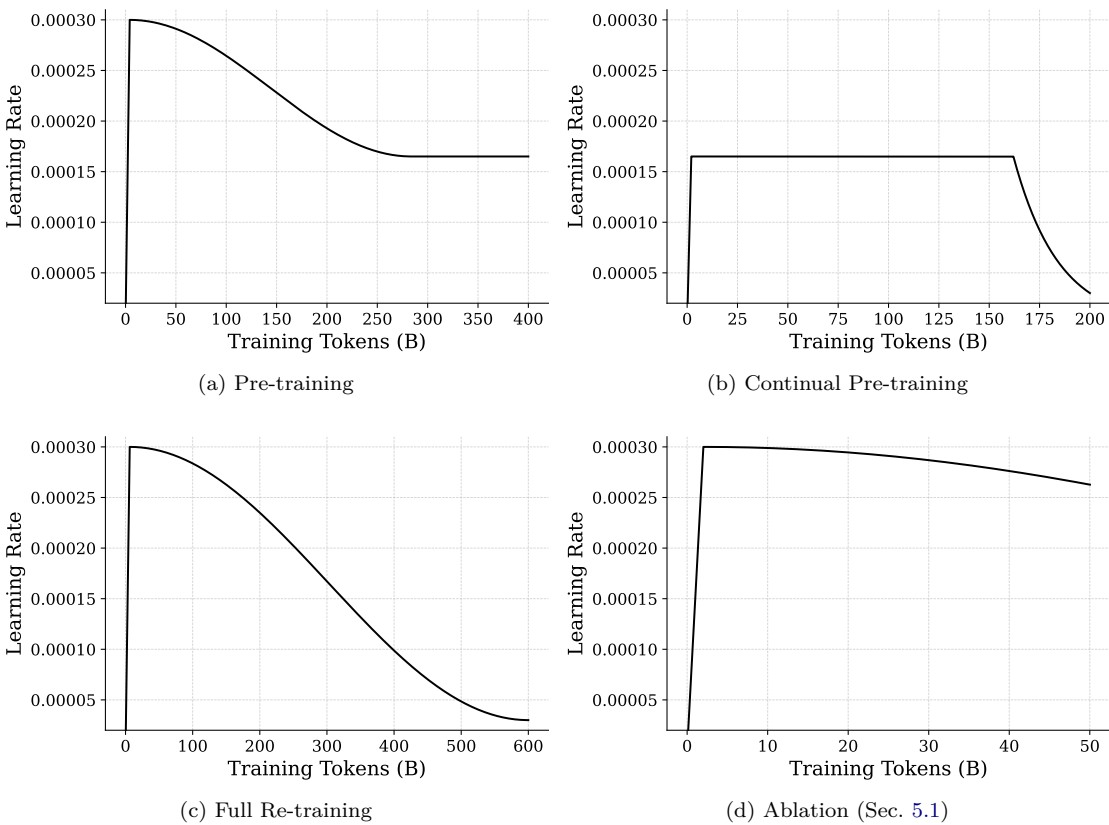

(a) Pre-training

(b) Continual Pre-training

(c) Full Re-training

(d) Ablation (Sec. 5.1)

Figure 22: **Illustrated Learning rate schedules used for (a) pre-training, (b) continual pre-training, (c) full re-training, and the (d) rewarming ablation of Sec. 5.1.** The exact hyperparameters of these schedules are reported in table 11.

Table 11: **Hyperparameters of LR schedules.** All models used the same LR schedule hyperparameters. We refer the readers to Ibrahim et al. (2024) section 7.2 for a more thorough explanation of these schedules.

| Description | Value |
|---|---|
| **Pre-training** | |
| Schedule Type | *CosineInf* |
| Total Iterations | 192720 |
| Max learning rate ($\eta_{max}$) | $3 \cdot 10^{-4}$ |
| Min learning rate ($\eta_{min}$) | $3 \cdot 10^{-5}$ |
| Constant learning rate ($\eta_{const}$) | $1.65 \cdot 10^{-4}$ |
| Warmup percent ($T_{warmup}$) | 1 |
| Cooldown iters percent ($T_{cd}$) | 70 |
| Constant iters percent ($T_{ann}$) | 0.10 |
| **Continual Pre-training** | |
| Schedule Type | *CosineInf* |
| Total Iterations | 95370 |
| Max learning rate ($\eta_{max}$) | $3 \cdot 10^{-4}$ |
| Min learning rate ($\eta_{min}$) | $3 \cdot 10^{-5}$ |
| Constant learning rate ($\eta_{const}$) | $1.65 \cdot 10^{-4}$ |
| Warmup percent ($T_{warmup}$) | 1 |
| Cooldown iters percent ($T_{cd}$) | 0 |
| Constant iters percent ($T_{ann}$) | 80 |
| **Full re-training** | |
| Schedule Type | *Cosine Annealing* |
| Total Iterations | 288090 |
| Max learning rate ($\eta_{max}$) | $3 \cdot 10^{-4}$ |
| Min learning rate ($\eta_{min}$) | $3 \cdot 10^{-5}$ |
| Warmup percent ($T_{warmup}$) | 1 |
| **Continual Pre-training Ablation** (Section 5.1) | |
| Schedule Type | *Cosine Annealing* |
| Total Iterations | 95370 |
| Max learning rate ($\eta_{max}$) | $3 \cdot 10^{-4}$ |
| Min learning rate ($\eta_{min}$) | $3 \cdot 10^{-5}$ |
| Warmup percent ($T_{warmup}$) | 1 |

Table 12: **Hyperparameters of our Moes and Dense Transformer.**

| Description | Value |
|---|---|
| **MoE Transformers Common** | |
| Active Parameters | $571, 148, 288$ |
| Parameters | $2, 025, 236, 480$ |
| Non-Embedding Parameters | $1, 893, 902, 336$ |
| **MoE SM-FFN** | |
| Shared Experts | 1 |
| Active Experts | 3 |
| Routed Experts | 31 |
| Total Experts | 32 |
| FFN Intermediate Size | 704 |
| **MoE R-FFN** | |
| Shared Experts | 0 |
| Active Experts | 1 |
| Routed Experts | 8 |
| Total Experts | 8 |
| FFN Intermediate Size | 2816 |
| **Top-$k$** | |
| Z-loss Coeff. | 0.001 |
| AUX-loss Coeff. | 0.01 |
| **Sinkhorn** | |
| Tolerance | 0.01 |
| **Dense Transformer** | |
| Parameters | $571, 148, 288$ |
| Non-Embedding Parameters | $439, 814, 144$ |
| Num attention heads | 16 |
| **Common** | |
| Num layers | 24 |
| Hidden size | 1024 |
| FFN Hidden size | 2816 |
| FFN Type | GeGLU |
| Optimizer | AdamW |
| $\beta_1, \beta_2$ | 0.9, 0.95 |
| Batch size | 1024 |
| Sequence length | 2048 |
| Hidden activation | GeLU |
| Weight decay | 0.1 |
| Gradient clipping | 1.0 |
| Decay | Cosine |
| Positional embedding | Rotary |
| GPT-J-Residual | True |
| Weight tying | False |
| Vocab Size | 128000 |
| Rotary PCT | 0.25 |

