# OpenReview forum: "Continual Pre-training of MoEs: How robust is your router?"
_TMLR — Accepted by TMLR_

### Review · Reviewer_HdyQ · 2025-08-10

**Summary Of Contributions:**

This paper presents a large-scale empirical study of CPT for MoE LLMs, comparing different MoE architectures and two routing strategies. The study also introduces Maximum Routing Imbalance as an evaluation of latency and load balance. The authors claim that MoEs maintain their sample-efficiency advantage during CPT, and are robust to distribution shift. They conclude that ghranular PBTk MoEs can match fully re-trained baselines at a fraction of the compute cost.

### Strengths
- The experimental design can be considered comprehensive, covering different routing algorithms and architectures with useful ablation studies. The results and analysis are clear and, from my perspective, reasonable.
- The topic covered by this article could be interesting and inspiring to researchers who are diving deep into MoE LLM fine-tuning design.

### Weaknesses
- The experiments are built upon a 570M model, which is somewhat small. Whether the conclusion can be extended to larger models is questionable.
- Due to the size of the model, down-stream evaluations are conducted on easy tasks.
- MRI is a proxy rather than direct latency measurements; no study of token-dropping or alternative routing beyond Top-k.

**Audience:**

Yes

**Audience Explanation:**

Yes. I think people will be interested in learning how MoE LLMs behave during fine-tuning.

**Claims And Evidence:**

Yes

**Claims Explanation:**

Yes. I think under the introduced setup, the claims are well-supported.

**Requested Changes:**

Although not necessary, I hope that the authors could include experiments on larger models, harder tasks, and probably model fine-tuning with quantization and more efficient approaches such as LoRA.

In addition, the authors should pay more attention to the paper format. For example, letter capitalization, citation format within the parentheses, etc.

---

> ### Author Response · Authors · 2025-08-26
> **Author reply part 1**
>
> Thank you for taking the time to review and engage with our work. We are pleased to hear that you believe our experiments are comprehensive and that our results analyses are clear and reasonable.
>
>
> **Why study MRI, instead of directly studying latency?** Our paper’s goal is to study the behaviour of MoEs during continual pre-training. Therefore, we made the decision to report behavioural metrics like MRI instead of system-level latency, which depends on a number of things (e.g., model size, interconnect speed, training precision, accelerator used, etc). While MRI does not report latency, it is a faithful behavioural metric that can be used as input to a latency model for estimating the latter. Unlike latency, which will always depend on specific hardware and implementation, MRI is independent of these considerations and is ultimately more comparable across different deployments. We have added a discussion of this to our section introducing MRI.
>
>
> **Request to train larger models and extrapolation of our results to larger scales**  Unfortunately, we are unable to run the study at a larger model scale due to computational restrictions. We believe that any empirical study will have the limitation that certain configurations were not tried due to computational budget, but we believe that our study still contains meaningful results and will be useful for practitioners. For instance, we chose to study 4 distinct MoE architectures representative of the current state of the art [3.3,3.4,3.7,3.8] with important baselines from the literature [3.5,3.6] across two realistic shifts: English→ Code and English → German. Although our results cannot reveal training difficulties that may arise at the 10B/100B+ model scale, we believe that many of the continual pre-training dynamics observed that are specific to the MoE architectures and routing algorithms we used will still be present at larger scales and can at least help inform such large-scale training runs.
>
> **No token dropping or routing methods beyond top-$k$** We did consider studying token dropping and routing algorithms beyond top-k, but since these techniques are not used in recent SOTA MoEs [3.3,3.4,3.7,3.8], we chose not to study them. As mentioned above, to maximize results within our available compute budget, we chose to study 4 distinct MoE architectures representative of the current state of the art [3.3,3.4,3.7,3.8] with important baselines from the literature [3.7,3.6]. We believe that any empirical study will have the limitation that certain configurations were not tried. However, given our choice of studying SOTA MoE architectures and routers, we believe that our results are representative of what is used in practice.
>
> **On the selection of downstream evaluations** As explained in section C.1 of the appendix, we decided to evaluate our models on downstream tasks that show above random chance accuracy. This decision was made to prevent noise caused by randomness in the evaluations from affecting our results. This has the side effect that we don’t evaluate on the hardest tasks since our models still obtain random chance at this scale.
>
>
> **Continual pre-training versus fine-tuning** Continual pre-training is distinct from fine-tuning due to the scale of new datasets. As we explain in section 2, within our paragraph about continual pre-training (CPT), CPT assumes that each new dataset is large (e.g., $>$100B tokens). The goal is to extend the pre-training phase to one or more new distributions. Continual pre-training can then be followed by fine-tuning phases, which typically use much less data.
>
> **Request to use LoRA** As we explained above, our focus is on continual pre-training, where an MoE transformer seeks to adapt to a large new dataset. Since methods like LoRA are designed for adapting to small-scale datasets, we chose not to include them. For example, in the LoRA paper [1] the authors train on GLUE, which consists of relatively modest benchmarks such as MNLI (393k examples), SST-2 (67k), and MRPC (3.7k), alongside others like CoLA (8.5k), QNLI (105k), QQP (364k), RTE (2.5k), and STS-B (7k)[2]. Other datasets used include: WikiSQL (56k training examples), SAMSum (14k), E2E (42k), DART (82k total), and WebNLG (~22k total). All have on the order of tens to hundreds of thousands of samples. Taking a liberal estimate of 4096 tokens per example (likely much larger than the reality), these datasets are still several orders of magnitude smaller than the hundreds of billions of tokens typical in continual pre-training.
>
> **Requested improvements to paper format** Please refer to our official comment where we have summarized the changes we made to the manuscript, including improving the letter capitalization and citation format.

---

> ### Author Response · Authors · 2025-08-26
> **Author reply part 2**
>
> **Local References**
> ---
> [1] [LORA: LOW-RANK ADAPTATION OF LARGE LANGUAGE MODELS; ICLR 2022]
>
> [2] [GLUE: A MULTI-TASK BENCHMARK AND ANALYSIS PLATFORM FOR NATURAL LANGUAGE UNDERSTANDING; ICLR 2019]
>
> [3.3] [DeepSeekMoE: Towards Ultimate Expert Specialization in Mixture-of-Experts Language Models]
>
> [3.4] [DeepSeek-V3 Technical Report]
>
> [3.5] [Switch Transformers: Scaling to Trillion Parameter Models with Simple and Efficient Sparsity; JMLR 2022]
>
> [3.6] [Unified Scaling Laws for Routed Language Models; ICML 2022 ]
>
> [3.7] [Qwen3 Technical Report]
>
> [3.8] [Kimi K2: Open Agentic Intelligence]

---

> > ### Comment · Reviewer_HdyQ · 2025-08-30
> >
> > Thank you for your responses! My concerns are well-addressed.

---

### Review · Reviewer_V6J6 · 2025-08-15

**Summary Of Contributions:**

The paper studies continual pre-training (CPT) for Mixture-of-Experts (MoE) LLMs and asks whether routers harm robustness under distribution shift. The authors consider four models (Granular vs Switch MoE; Penalty-Balanced Top-k (PBTk) VS Sinkhorn-Balanced Top-k (SBTk)), and pre-train them on 400B FineWeb tokens, then continue on 200B tokens of German or The Stack with replay and (usually) an infinite-LR schedule from a non-decayed checkpoint.

In Section 4, the paper provides an ablation study; increasing replay reduces forgetting on the original domain but hurts adaptation to the new one (which is quite trivial as replay decreases the amount of tokens from new distribution), and starting CPT from a non-decayed checkpoint (infinite LR) improves forgetting. Importantly, they introduce Maximum Routing Imbalance (MRI), the maximum share of tokens sent to any single expert in a layer, and use it as a proxy for worst-case MoE latency. In Section 5.1, the authors observe that SBTk is very stable and shows only a small MRI bump, while PBTk is unstable right after the shift, but quickly recovers to a lower MRI than SBTk. In Section 5.2, all MoEs beat the FLOP-matched dense baseline on validation loss during both pre-training and CPT; within MoEs, PBTk consistently outperforms SBTk and Granular outperforms Switch. Also, CPT MoEs match the full re-training baseline on downstream evaluations, indicating that routers do not block CPT and that MoEs keep their sample-efficiency benefit. In Section 5.3, the authors analyze how routing changes: the authors adapt Router Saturation, Vocabulary Specialization, and Expert Co-activation to the CPT setting.

**Additional Comments:**

N/A

**Audience:**

Yes

**Audience Explanation:**

MoE LLMs are a major trend right now (e.g., DeepSeek, Kimi K2). I think that decoder-only MoE Transformers differ in important ways from standard dense models, so checking whether classic CPT techniques still work, and how routing behaves under shift is a nontrivial and important question. While the paper does not introduce a new method or a striking new phenomenon, it provides a detailed systematic understanding of existing practices: the benefits of replay and starting from non-decayed checkpoint, and a clear comparison of routing (PBTk vs. SBTk). In that sense, I consider the findings valuable, and TMLR's audience would be interested in this paper as well.

**Broader Impact Concerns:**

The reviewer does not identify any ethical concerns.

**Claims And Evidence:**

Yes

**Claims Explanation:**

I believe most of the claims are supported by clear experimental evidence, but a few seem insufficiently justified.

In section 5.1 (routing imbalance), the authors claim that PBTk shows a large imbalance at the very beginning of CPT but becomes more balanced than SBTk as training continues, and that this could be a factor behind PBTk’s better validation loss in Figure 2. However, I do not see a clear connection between the balancing metric (MRI) and validation loss, so this claim feels insufficiently justified.

Also, there is a confusing point regarding the MRI-related plots in this paper. In Figure 1, the MRI scale appears to range from about 3 to 6, whereas in Figure 3 it stays below 0.4, and in Figure 4 it is presented as a percentage. A detailed explanation is needed to clarify exactly what values or normalizations were used in each figure.

**Requested Changes:**

The topic, claims, and direction are good, but the paper feels incomplete at the current stage in terms of presentation. Here I list few of them:

### Typos/Mistakes

- Figure 1: “400B English → 200B German” appears twice; one of them should be “→ 200B Stack.”
- p.4: “FineWeb Penedo et al. (2024)” → “FineWeb (Penedo et al., 2024)”
- p.4: “chinchilla optional” → “chinchilla optimal”
- p.6: “disribution” → “distribution”
- p.10: “models.Our” → “models. Our”
- p.22: “Cotinual” → “Continual”
- Unify capitalization style for appendix subtitles.

### Lack of Description / Presentation Gaps
- (Figure 1) I may have missed it, but I don’t see which exact setting the results come from, especially the replay ratio. Please specify.
- The main text under-explains what is deferred to the appendix. For example, in Section 4 when describing the method, add explicit pointers like “see Appendix E for full hyperparameters.”
- (Figure 3) (a) and (b) use purple = SBTk and black = PBTk, but panel (c) swaps them.

---

> ### Author Response · Authors · 2025-08-26
> **Author reply**
>
> Thank you for taking the time to review and engage with our work. We are particularly pleased to hear that you believe our work studies a non-trivial and important question and that you believe our findings are valuable.
>
> **Claim linking routing imbalance recovery and performance** Thanks for pointing this out. We have updated the claim to: (1) make it clear that we are speculating about a link between SB MRI and validation loss and (2) to be more precise about how the two may be linked. The text currently reads:
>
> > These results suggest that although SB is more robust to distribution shifts than PB, this robustness limits the MRI attainable and could be the cause of the poorer performance seen above for validation loss.
>
> this now becomes:
>
> > These results suggest that although SB is more robust to distribution shifts than PB, this robustness limits the MRI attainable. We hypothesize that the generally higher MRI of PB models may cause an uneven utilization of the MoE’s parameters during training. Such a difference in expert utilization during training could possibly explain the differences in performance between PB and SB.
>
>
>
> **A detailed description of how MRI is used in Figures 3 and 4** Thank you for pointing this out. We have updated Figure 3 and the corresponding figures in the appendix to also use a percentage. The difference between MRI reported in 3 and 4 is that in 3, we report the maximum MRI across all layers at each training step, while Figure 4 shows the layer-wise MRI of final checkpoints. We have included a discussion of the differences in usage between each plot at the end of section 4.4.
>
> **Adding setting to figure 1** Thank you for pointing this out, we have now updated the caption with more information.
>
> **Including references to the appendix in the main text** Thank you for pointing this out, we have mentioned that hyperparameter descriptions are deferred to the appendix.
>
> **Figure 3 colors swapped** Thank you for pointing this out. We have fixed this.
>
> **Minor Modification** Please refer to our official comment where we have summarized the changes we made to the manuscript.

---

> > ### Comment · Reviewer_V6J6 · 2025-08-31
> >
> > Thank you for your response. I am now satisfied with your argument on routing imbalance recovery.
> >
> > When I first reviewed this paper, I felt it was a bit rough in terms of polish. Besides the specific errors I pointed out, I hope the authors carefully go over the paper again to finish it well.

---

### Review · Reviewer_Ghyb · 2025-08-18

**Summary Of Contributions:**

This paper investigates how Mixture of Expert (MoE) transformer LLMs behave during continued pre-training (CPT) when exposed to distribution shifts. The assumption is that CPT might be challenging for MoEs because the router might not be able to deal with distribution shifts well - in terms of performance and efficiency (load balancing, etc). The authors train a ~500M parameter dense model and isoFLOP MoE models of different variants on FineWeb and do CPT on German language and code. They present insightful ablations, analyses and clear practical guidelines.

**Additional Comments:**

Additional questions:
1. Why is Stack 0% missing from Figure 5?
2. When using replay, does that increase the total CPT token budget? E.g. using 0% replay with 200B tokens, will using 40% replay use 280B tokens?
3. Why use different replay percentages for German (40%) vs Stack (30%)?

**Audience:**

Yes

**Audience Explanation:**

I believe that this work presents a good foundation for future work to build on. Readers will be interested in learning how CPT works for MoE models and what new tradeoffs MoEs present in this setting. The authors nicely point to future directions of inquiry.

**Broader Impact Concerns:**

No concerns.

**Claims And Evidence:**

Yes

**Claims Explanation:**

Yes, but I believe two points are not quite clear currently.
1. The models are very overtrained at 40x Chinchilla optimal for the dense one. It's not clear that the results will translate to compute optimal regimes (which is what's most relevant for practical applications).
2. All comparisons between MoE and Dense are biased towards the dense model. The comparison is MoE vs. FLOP-matched Dense which is not fair (see [Du et al. 2024](https://arxiv.org/abs/2405.15052v2)) as the MoE will be slower due to higher memory and communication requirements. See also point 2 in requested changes below

**Requested Changes:**

1. Regarding point 1 above, please mention the dense model size in the abstract in addition to the number of tokens which is already mentioned. Right now the text mentions "large-scale (>2B parameter switch and DeepSeek MoE LLMs trained for 600B tokens)" but this is misleading because DeepSeek MoE models are not actually used, and the meaning of ">2B parameter switch" is not clear from the context.
2. It would be very insightful to add the time per training/inference step for the reader to see a full comparison of efficiency for dense vs. MoE models. This would be a much better proxy for efficiency than MRI. At the very least it would be beneficial to report latency measurements for the different models that are investigated.

Minor:
1. MRI is a only a proxy for worst-case latency, not latency (change title of section 4.4)
2. Citation brackets need fixing, use `Authors (2024)` vs. `(Authors, 2024)` correctly.
3. Typo in title of Appendix B.2
4. Please clarify the explanation of the Vocabulary Specialization in Section 5.3: "...calculate the average VS of each expert by averaging over its assigned tokens and..." - what do you average over tokens? Surely not the token ID. Do you take the count of tokens per expert?
5. The plots in Figure 22 are slightly confusing with the text in Section 4.1. Which of the plots show the CosineInf schedule? Does setting (b) in 5.1 start with a warmup or not? Please clarify.

---

> ### Author Response · Authors · 2025-08-26
> **Author reply part 1**
>
> Thank you for taking the time to review and engage with our work. We are pleased that you believe our work presents a good foundation for future work in the context of MoE Continual pre-training and that our work nicely points to future directions of inquiry.
>
> **Compute Optimality in a Continual pre-training setting** While we agree with the reviewer that compute optimality is important to large-scale pre-training, we disagree that starting from a compute-optimally trained model is most relevant for practical applications. For example, the extremely popular Qwen3 series of models were each pre-trained for 36T tokens [1.1]. This corresponds to a chinchilla optimal multiplier of 7.66-58.06 for MoEs and 54.55-225 for larger dense models (see table below). Note that these models are frequently used as a starting point for continual pre-training, showing how our 40x chinchilla-optimal for dense and 10x chinchilla-optimal for MoEs is representative of real application settings. We have added the discussion of the Qwen 3 models to section 4.2.
>
>
> | Model Version   | Parameters (B) | Training Tokens | Chinchilla Multiplier | HuggingFace Downloads (last month) |
> |-----------------|----------------|-----------------|------------------------|-----------|
> | Qwen3-235B-A22B | 235            | 36T             | 7.66                   | 141,000   |
> | Qwen3-30B-A3B   | 31             | 36T             | 58.06                  | 1,100,000 |
> | Qwen3-32B       | 33             | 36T             | 54.55                  | 993,000   |
> | Qwen3-14B       | 15             | 36T             | 120.00                 | 959,000   |
> | Qwen3-8B        | 8              | 36T             | 225.00                 | 4,990,000 |
>
> **Latency v.s. MRI** Our paper’s goal is to study the behaviour of MoEs during continual pre-training. Therefore, we made the decision to report behavioural metrics like MRI instead of system-level latency, which depends on a number of things (e.g., model size, interconnect speed, training precision, accelerator used, etc). While MRI does not report latency, it is a faithful behavioural metric that can be used as input to a latency model for estimating the latter. Unlike latency, which will always depend on specific hardware and implementation, MRI is independent of these considerations and is ultimately more comparable across different deployments. We have added a discussion of this to our section introducing MRI.
>
> **Comparisons are biased towards the MoEs** We acknowledge that there are non-negligible downsides to training MoEs compared to a FLOP-matched dense model, including higher memory, longer optimizer step, and higher communication costs. However, we would like to emphasize that our goal is to study MoEs during continual pre-training and that we make no claims about the communication efficiency, memory efficiency, or optimizer step times of MoEs compared to dense models. We have integrated your reference about the MoE speed-accuracy tradeoff [1.2] within our new section about model timings. Thank you for bringing it to our attention.
>
> **Adding Step time of MoEs compared to dense models** We have added a table reporting the training step time of each MoE architecture in our study compared to the Dense models in the appendix. We have also noted that the step times we report are dependent on the  GroupedGEMM kernel of [1.3], which was used when training MoEs in our study. We also make clear to the reader that the latency of and MoE’s forward pass depends on many factors:  model size, interconnect speed, training precision, accelerator used, etc.
>
> **Modification to the abstract** Thank you for pointing out the ambiguity here. We have made the following modification to our abstract, the existing text:
>
> > In what follows, we conduct a large-scale (> 2B parameter switch and DeepSeek MoE LLMs trained for 600B tokens) empirical study across four MoE transformers to answer these questions.
>
> Now becomes:
>
> >  In what follows, we conduct a large-scale study training a 500M parameter dense transformer and four 500M-active/2B-total parameter MoE transformers, following the Switch Transformer architecture and a granular DeepSeek-inspired architecture. Each model is trained for 600B tokens.
>
> **Why is Stack 0% missing from Figure 5?** To reduce experimental cost, we decided to run all of our experiments, which ablate replay, on the German dataset.
>
> **Does replay increase the compute budget?** As mentioned in section A.1 of the appendix, we use compute equivalent replay, meaning that increasing the replay percentage decreases the number of new tokens seen, but keeps FLOPs constant. We have moved this explanation to the main text.

---

> ### Author Response · Authors · 2025-08-26
> **Author reply Part 2**
>
> **Replay percentage selection** For code, we chose to use 30% replay since the value was used in previous SOTA work [1.4]. For German, we took inspiration from [1.5, Table 2], which showed non-negligible differences in forgetting between 25% replay and 50% replay in the Dense model CPT setting. To help prevent forgetting more than the 25% replay model of [1.5, Table 2] but without degrading adaptation as much as the 50% replay model in [1.5, Table 2], we selected a value in between: 40%.
>
> **Minor Modification** Please refer to our official comment where we have summarized the changes we made to the manuscript.
>
> **Local References**
> ---
> [1.1][Qwen3 Technical Report]
>
> [1.2][Revisiting MoE and Dense Speed-Accuracy Comparisons for LLM Training]
>
> [1.3][MegaBlocks: Efficient Sparse Training with Mixture-of-Experts]
>
> [1.4][DeepSeek-Coder-V2: Breaking the Barrier of Closed-Source Models in Code Intelligence]
>
> [1.5 [Simple and Scalable Strategies to Continually Pre-train Large Language Model; TMLR 2024]

---

### Author Response · Authors · 2025-08-26
**Revision 1 based on reviewer's initial feedback**

Dear reviewers,

We would like to thank you for taking the time to review our paper.

In light of your feedback, we have made the following modifications to our paper. To make changes to the manuscript easier to read, we have highlighted any changes to a sentence's wording or its meaning in blue. This excludes small changes to fix typos.



| Reviewer | Description of Change | Status |
|----------|------------------------|--------|
| Bhyb | Change title of Section 4.4: "MRI is only a proxy for worst-case latency, not latency" | X |
| Bhyb | Fix citation brackets to use *Authors (2024)* vs. *(Authors, 2024)* correctly | X |
| Bhyb | Correct typo in title of Appendix B.2 | X |
| Bhyb | Clarify explanation of Vocabulary Specialization in Section 5.3: specify what is averaged over tokens (not token IDs, but counts per expert) | X |
| Bhyb | Clarify alignment between Figure 22 plots and Section 4.1 text (indicate which plots correspond to which explanations) |X |
| Bhyb | Add information about Qwen MoEs and compute the optimal budget in Section 4.2 | X |
| Bhyb | Move compute equivalent replay explanation to the main text | X |
| Bhyb | Modify the abstract to specify the dense model size used and MoE architectures | X |
| Bhyb | Add training timings for the different MoEs in our study to the appendix with appropriate caveats | X |
| V6J6 | Fix Figure 1 label: “400B English → 200B German” appears twice; correct one to “→ 200B Stack” | X |
| V6J6 | Correct citation formatting: “FineWeb Penedo et al. (2024)” → “FineWeb (Penedo et al., 2024)” | X |
| V6J6 | Fix typo: “chinchilla optional” → “chinchilla optimal” | X |
| V6J6 | Fix typo: “disribution” → “distribution” | X |
| V6J6 | Fix typo: “models.Our” → “models. Our” | X |
| V6J6 | Fix typo: “Cotinual” → “Continual” | X |
| V6J6 | Unify capitalization style for appendix subtitles |X |
| V6J6 | Figure 1: specify exact experimental setting, especially replay ratio | X|
| V6J6 | Section 4: add explicit pointers to appendix (e.g., “see Appendix E for full hyperparameters”) | X |
| V6J6 | Figure 3: fix inconsistent color mapping (a,b: purple = SBTk, black = PBTk; c swaps them) | X |
| V6J6 | Ensure Figure 3 presentation follows percentage style, consistent with Figure 4 | X |
| V6J6 | Include discussion of differences in MRI usage across plots at the end of Section 4.4 | X|
| HdyQ | Add a discussion of MRI vs latency in the section about MRI | X|
| HdyQ | Improve paper format (general polishing and presentation) | X |

---

### Decision · Action_Editor_NTXn · 2025-09-18

**Recommendation:** Accept as is

**Audience:**

Yes

**Audience Explanation:**

These results have substantial practical interest to those who wish to adapt pretrained models to new data without having to retrain from scratch.

**Claims And Evidence:**

Yes

**Claims Explanation:**

The reviewers and I agree that this paper provides a useful and accurate analysis of how MoE routing affects continual pretraining (relative to dense models). The claims are supported by the evidence, especially after the revision.